# PDPO: Parametric Density Path Optimization

**Sebastian Gutierrez Hernandez**[1]    **Peng Chen**[2]    **Haomin Zhou**[1]

[1]School of Mathematics, Georgia Institute of Technology
[2]School of Computational Science and Engineering, Georgia Institute of Technology

{shern3, pchen402 ,hmzhou}@gatech.edu

## Abstract

We introduce Parametric Density Path Optimization (PDPO), a novel method for computing action-minimizing paths between probability densities. The core idea is to represent the target probability path as the pushforward of a reference density through a parametric map, transforming the original infinite-dimensional optimization over densities to a finite-dimensional one over the parameters of the map. We derive a static formulation of the dynamic problem of action minimization and propose cubic spline interpolation of the path in parameter space to solve the static problem. Theoretically, we establish an error bound of the action under proper assumptions on the regularity of the parameter path. Empirically, we find that using 3–5 control points of the spline interpolation suffices to accurately resolve both multimodal and high-dimensional problems. We demonstrate that PDPO can flexibly accommodate a wide range of potential terms, including those modeling obstacles, mean-field interactions, stochastic control, and higher-order dynamics. Our method outperforms existing state-of-the-art approaches in benchmark tasks, demonstrating superior computational efficiency and solution quality. Source code `https://github.com/SebasGutHdz/PDPO/tree/main` .

## 1   Introduction

Optimal transport (OT) theory has become a powerful and widely used framework for quantifying discrepancies and constructing interpolations between probability distributions, with rapidly expanding applications in machine learning, statistics, and computational science [27]. It offers a principled way to compare distributions by computing the minimal cost required to transform one distribution into another.

While the computation of optimal transport (OT) plans and geodesics between probability distributions is now relatively well understood [31, 22, 3, 34, 25, 13], many real-world applications require solving more complex, constrained transport problems—specifically, identifying optimal paths between distributions subject to additional requirements. For instance, one may need to guide a swarm of robots from one configuration to another while avoiding obstacles, or interpolate between data distributions in a way that preserves semantic structure [20, 29, 17].

These tasks can be formulated as action-minimizing problems governed by the "principle of least action": Given initial and terminal densities $\rho_0, \rho_1 \in \mathcal{P}(\mathbb{R}^d)$, where $\mathcal{P}(\mathbb{R}^d)$ denotes the space of probability densities over $\mathbb{R}^d$, the objective is to find a time-dependent density path $\rho(t, \cdot) \in \mathcal{P}(\mathbb{R}^d)$ and a continuous velocity field $v : [0, 1] \times \mathbb{R}^d \to \mathbb{R}^d$ that minimize the action functional:

$$\inf_{\rho, v} \int_0^1 \int_{\mathbb{R}^d} K(\rho, v) \, dx + F(\rho(t)) \, dt \tag{1}$$

$$\text{subject to} \quad \partial_t \rho + \nabla \cdot (\rho v) = 0, \quad \rho(0, \cdot) = \rho_0, \quad \rho(1, \cdot) = \rho_1. \tag{2}$$

39th Conference on Neural Information Processing Systems (NeurIPS 2025).

Here, $K(\rho, v)$ denotes the transportation energy, and $F(\rho)$ represents a potential term that captures interactions among particles or with the environment. When $K(\rho, v) = \frac{1}{2}|v|^2 \rho$ and $F(\rho) = 0$, this reduces to the classical Wasserstein geodesic problem. Directly solving such action-minimizing problems poses substantial mathematical and computational challenges. In low dimensions (e.g., 2 or 3), the associated PDE system—derived from the first-order optimality conditions—can be tackled using classical numerical methods [1, 5, 6, 14]. However, these approaches do not scale well with dimension and become computationally infeasible in high-dimensional settings.

Recent advances in machine learning have greatly expanded the range of high-dimensional problems that can be tackled effectively. A leading example is the Generalized Schrödinger Bridge Method (GSBM) [20], originally developed for stochastic optimal control (SOC) problems [24]. GSBM learns forward and backward vector fields by modeling conditional densities and velocities, drawing inspiration from stochastic interpolants [2], and approximates Gaussian path statistics via spline-based optimization. In parallel, [29] introduced an algorithm for problems with linear energy potentials, leveraging Kantorovich duality and amortized inference to efficiently compute c-transforms and transportation costs. For Mean-Field Games, APAC-Net [17] casts the primal-dual formulation as a convex-concave saddle point problem, trained using a GAN-style adversarial framework.

Our method builds upon prior work on parametric probability distributions [21, 33, 15], extending these ideas to address boundary-valued action-minimizing density problems. Our formulation is an extension of the *static OT* formulation to the parameter space, whereas in [16], they extend the *dynamic formulation*. In the remainder of this section, we outline our methodology, with full technical details deferred to Section 3.

Figure 1a offers an overview of our approach: curves in parameter space $\theta(t) \in \Theta$ (left) induce density paths $\rho_t \in \mathcal{P}(\mathbb{R}^d)$ via pushforward maps $\rho_t = (T_{\theta(t)})_\# \lambda$ from a fixed reference density $\lambda$ (top), while samples of these densities are obtained via direct evaluation $T_{\theta(t)}(z)$ with $z \sim \lambda$ (right).

While spline-based methods have previously been applied to particle-level density transport [29] and Gaussian settings [20], a key innovation of our work is the use of cubic Hermite splines in the parameter space of a neural network. We fix a set of control points $\{\theta_{t_i}\}_{i=0}^{K+1}$ with $t_i = i/(K+1)$ and optimize over them to discover action-minimizing density paths, which presents the first spline-based parametric approach in a learned map setting. Figures 1b and 1c illustrate our method on an obstacle avoidance task, showing the pushforward of $\lambda$ at the optimized control points and the resulting continuous path of probability densities.

The optimization of the $K + 2$ control points proceeds in the following steps: **Step 1:** Initialize a pair of boundary parameters, $\theta_0$ and $\theta_1$, that accurately approximate the target boundary densities

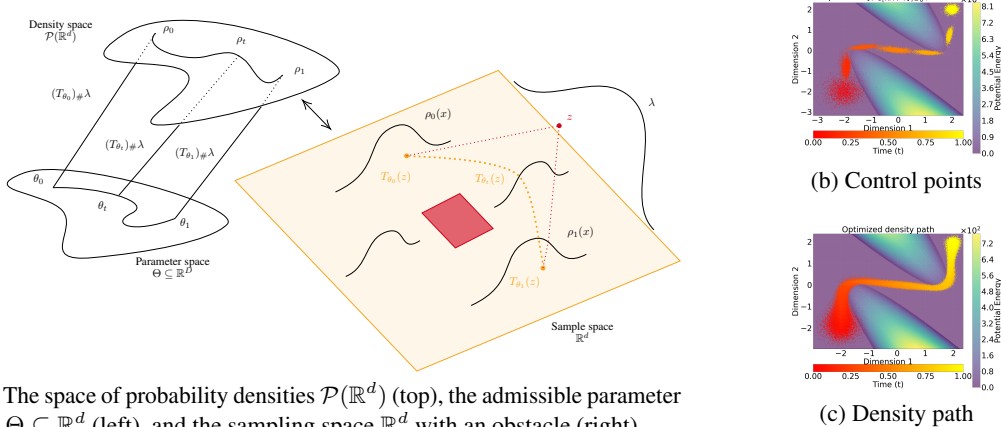

(a) The space of probability densities $\mathcal{P}(\mathbb{R}^d)$ (top), the admissible parameter set $\Theta \subseteq \mathbb{R}^d$ (left), and the sampling space $\mathbb{R}^d$ with an obstacle (right)

(b) Control points

(c) Density path

Figure 1: Visualization of our framework. (a) Illustrates the three main components: the space of probability densities $\mathcal{P}(\mathbb{R}^d)$ (top), the admissible parameter space $\Theta \subseteq \mathbb{R}^d$ (left), and the sampling space $\mathbb{R}^d$ containing an obstacle (right). (b) and (c) depict the pushforwards of the optimized control points $(T_{\theta_i})_\# \lambda$ for $i = 0, \ldots, K$ and the continuous spline trajectory $(T_{\theta(t)})_\# \lambda$, respectively. Time is color-coded, with red indicating $t = 0$ and yellow indicating $t = 1$.

$\rho_0$ and $\rho_1$. **Step 2:** With the boundary parameters $(\theta_0, \theta_1)$ fixed, optimize the interior control points $\{\theta_{t_i}\}_{i=1}^{K+1}$ to minimize the action associated with the coupling $(T_{\theta_0}, T_{\theta_1})_{\#}\lambda$. **Step 3:** Optimize the boundary parameters $(\theta_0, \theta_1)$ in order to improve the quality of the coupling while keeping the interior control points fixed. Repeat steps 2 and 3 iteratively until convergence.

**Contributions**. Our main contributions are: (1) We introduce Parametric Density Path Optimization (PDPO), a novel framework that transforms infinite-dimensional density path optimization into a finite-dimensional problem by parameterizing pushforward maps with cubic Hermite splines. Our approach achieves accurate density paths with as few as 3–5 control points. (2) We generalize the parametric framework to support a wide class of action-minimizing problems, including stochastic optimal control (via Fisher information), acceleration-regularized dynamics, and mean-field interactions, within a unified optimization scheme based on learned transport maps. (3) We empirically validate PDPO across several challenging benchmarks—including obstacle avoidance, entropy-constrained transport, and high-dimensional opinion dynamics—achieving up to 7% lower action, up to $10\times$ improved boundary accuracy, and 40–80% faster runtimes compared to state-of-the-art baselines.

## 2 Background

This section presents the mathematical foundations of optimal transport, its dynamic formulation, and parametric methods that enable scalable and efficient application in machine learning contexts.

### 2.1 Optimal transport and Wasserstein distances

In what follows, we only consider distributions $\mu$ that are Lebesgue dominated, thus there is a function $\rho : \mathbb{R}^d \to \mathbb{R}_{\geq 0}$ such that for any measurable set $A$, $\mu(A) = \int_A \rho(x)dx$. Since we are restricting ourselves to this case, we use the terms distribution and density interchangeably for convenience.

The Wasserstein-2 distance between two probability distributions $\rho_0, \rho_1 \in \mathcal{P}(\mathbb{R}^d)$ is defined as:

$$W_2^2(\rho_0, \rho_1) = \inf_{\pi \in \Pi(\rho_0, \rho_1)} \int_{\mathbb{R}^d \times \mathbb{R}^d} \|x - y\|^2 d\pi(x, y), \tag{3}$$

where $\Pi(\rho_0, \rho_1)$ denotes the set of all joint distributions with marginals $\rho_0$ and $\rho_1$ [32].

When $\rho_0$ is absolutely continuous with respect to the Lebesgue measure, the Benamou–Brenier formulation [4] provides a dynamic reinterpretation of this distance:

$$W_2^2(\rho_0, \rho_1) = \inf_{\rho, v} \int_0^1 \int_{\mathbb{R}^d} \frac{1}{2} \|v(t, x)\|^2 \rho(t, x) dx dt \tag{4}$$

$$\text{subject to} \quad \partial_t \rho + \nabla \cdot (\rho v) = 0, \quad \rho(0, \cdot) = \rho_0, \quad \rho(1, \cdot) = \rho_1. \tag{5}$$

This dynamical formulation interprets the Wasserstein distance as the minimum kinetic energy required to continuously transport the mass of $\rho_0$ into $\rho_1$ over the time interval $[0, 1]$.

### 2.2 Generalized action-minimizing problems in Wasserstein space

We consider the following generalized action-minimizing problem:

$$\inf_{\rho, v} \int_0^1 \int_{\mathbb{R}^d} \frac{1}{2} \|v(t, x)\|^2 \rho(t, x) dx + F(\rho(t)) \, dt \tag{6}$$

$$\text{subject to} \quad \partial_t \rho + \nabla \cdot (\rho v) = 0, \quad \rho(0, \cdot) = \rho_0, \quad \rho(1, \cdot) = \rho_1,$$

where the functional $F(\rho)$ incorporates additional constraints or potentials, defined as:

$$F(\rho) = \kappa_0 \int_{\mathbb{R}^d} V(x)\rho(x)dx + \kappa_1 \int_{\mathbb{R}^d} U(\rho(x))dx + \kappa_2 \int_{\mathbb{R}^d \times \mathbb{R}^d} W(x - y)\rho(x)\rho(y)dxdy. \tag{7}$$

Here, the three terms correspond respectively to: external potentials (e.g., obstacles or environmental forces), internal energy (e.g., entropy or Fisher information), and interaction energy (e.g., repulsion or attraction between mass elements). The constants $\kappa_0, \kappa_1, \kappa_2 \in \mathbb{R}_+$ model the strength of each term.

In our setup, the Fisher Information potential, given by

$$\mathcal{FI}(\rho) = \frac{\sigma^4}{8} \int \|\nabla \log \rho\|^2 \rho(x) dx, \tag{8}$$

plays a crucial role. As shown in prior work [11, 23] and detailed in Appendix A, it is well established that the following deterministic control problem:

$$\inf_{\rho,v} \int_0^1 \int_{\mathbb{R}^d} \frac{1}{2} \|v(t,x)\|^2 \rho(t,x) dx + F(\rho(t)) + \mathcal{FI}(\rho(t)) \, dt$$
$$\text{subject to} \quad \partial_t \rho + \nabla \cdot (\rho v) = 0, \quad \rho(0, \cdot) = \rho_0, \quad \rho(1, \cdot) = \rho_1,$$

is equivalent to the stochastic optimal control (SOC) problem:

$$\inf_{\rho,v} \int_0^1 \int_{\mathbb{R}^d} \frac{1}{2} \|v(t,x)\|^2 \rho(t,x) dx + F(\rho(t)) \, dt$$
$$\text{subject to} \quad \partial_t \rho + \nabla \cdot (\rho v) = \frac{\sigma^2}{2} \Delta \rho, \quad \rho(0, \cdot) = \rho_0, \quad \rho(1, \cdot) = \rho_1.$$

This equivalence allows us to recast SOC problems as deterministic action-minimization tasks augmented with a Fisher Information regularization term. While the deterministic action-minimization formulation offers a principled framework for modeling action-minimizing dynamics, directly solving the resulting infinite-dimensional optimization problem over probability densities is computationally prohibitive in high dimensions. To address this, we next introduce parametric representations that reduce the problem to a finite-dimensional setting, enabling tractable and scalable computation.

## 2.3 Parametric pushforward representations of probability densities

To render action-minimizing problems computationally tractable, we focus on parameterized families of probability distributions, thereby reducing the original infinite-dimensional optimization problem to a finite-dimensional one over the parameter space. This strategy builds on recent developments in parameterized Wasserstein dynamics for solving initial-value Hamiltonian systems and gradient flows [33, 15].

The work most closely related to our approach is that of [8], which formulates the Schrödinger Bridge (SB) problem between Gaussian distributions as an action-minimization task. Leveraging the fact that the SB solution between Gaussians remains Gaussian, they parameterize the evolving density by its mean and covariance, and derive Euler–Lagrange equations for these parameters by projecting the density-level optimality conditions onto the parametric space.

We now introduce the core elements of our parametric pushforward formulation, starting with the definition of some key concepts. For further background, we refer the reader to [33, 16].

**Definition 1.** *We consider a **parameter space** $\Theta \subseteq \mathbb{R}^D$ of dimension $D$. We call $T : \mathbb{R}^d \times \Theta \to \mathbb{R}^d$ a **parametric mapping**, if for every $\theta \in \Theta$, the function $T(\cdot, \theta)$, or $T_\theta(\cdot)$ used interchangeably, is Lebesgue measurable. The corresponding **parametric density space** is defined as*

$$P_\Theta := \{\rho_\theta = (T_\theta)_{\#}\lambda : \theta \in \Theta\} \subseteq \mathcal{P}(\mathbb{R}^d),$$

*where $(T_\theta)_{\#}\lambda$ denotes the pushforward of a fixed reference density $\lambda$ under the map $T_\theta$, see Figure 1a(top). Moreover, given a curve in the parameter space $\theta(t) \in \Theta$ indicated by $t \in [0, 1]$, the corresponding **pushforward density path** is defined as $\rho_{\theta(t)} := (T_{\theta(t)})_{\#}\lambda$, see Figure 1a(top), and the **sample trajectories** are given by $x_{\theta(t)}(z) = T_{\theta(t)}(z), z \sim \lambda$, see Figure 1a(right).*

# 3 Action Minimizing Problems via Parametric Pushforwards

## 3.1 Projection of the action-minimizing problem to the parametric setting

To motivate our framework, we begin by revisiting an alternative formulation of the classical Wasserstein-2 distance defined in (3). Given $x, y \in \mathbb{R}^d$, define the set of admissible paths $\Gamma(x, y) := \{\gamma(t) \in C^1([0, 1], \mathbb{R}^d) : \gamma(0) = x, \gamma(1) = y\}$. Then the classical formula

$$\|x - y\|^2 = \inf_{\gamma(t) \in \Gamma(x,y)} \int_0^1 \|\gamma'(t)\|^2 dt,$$

allows us to write the Wasserstein-2 distance as

$$W_2^2(\rho_0, \rho_1) = \inf_{\pi \in \Pi(\rho_0, \rho_1)} \int_{\mathbb{R}^d \times \mathbb{R}^d} \inf_{\gamma(t) \in \Gamma(x,y)} \int_0^1 \|\gamma'(t)\|^2 \, dt \, d\pi(x,y). \tag{9}$$

In other words, one first chooses a coupling $\pi$ between the endpoints and then pays, for each pair $(x,y)$, the minimal kinetic energy needed to drive a particle from $x$ to $y$. Our framework adopts exactly this viewpoint: we cast the dynamic optimization in (6) as a static coupling problem whose cost is still evaluated via a dynamic (least-action) criterion.

For generalized action-minimizing problems, the action of a path is

$$\mathcal{A}(\gamma) = \mathbb{E}_{x \sim \rho_0} \left[ \int_0^1 \left\| \frac{d}{dt} \gamma_t(x) \right\|^2 dt \right] + \int_0^1 F(\rho_{\gamma_t}) dt. \tag{10}$$

We make the following restrictions to guarantee the existence of the density in all the trajectories of a coupling. Define the set

$$\mathcal{T}_{\rho_0, \rho_1} = \{ T : \mathbb{R}^d \to \mathbb{R}^d : T \text{ diffeomorphic}, T_\#(\rho_0) = \rho_1, \det(DT) > 0 \}.$$

Since the $\rho_0$ and $\rho_1$ have densities, the Monge map $T^*$ is a diffeomorphism that satisfies $\det(DT) > 0$ [4] and the set $\mathcal{T}_{\rho_0, \rho_1} \neq \emptyset$. If $T \in \mathcal{T}_{\rho_0, \rho_1}$ then it is smoothly isotopic to the identity $id$, see Lemma 2, Chapter 6 [7]. By the definition of diffeomorphism being smoothly isotopic, for maps $T \in \mathcal{T}_{\rho_0, \rho_1}$, the set $\Gamma(\pi)$ consisting of the flow maps $\gamma \in C^1([0,1] \times \mathbb{R}^d, \mathbb{R}^d)$, such that $\gamma_t(x) := \gamma(t,x)$ describes the position at time $t$ of a particle starting at $x$, i.e., $\gamma_0(x) = x$, $\gamma_t(\cdot)$ is diffeomorphic for each $t \in [0,1]$, and $(\gamma_0, \gamma_1)_\# \rho_0 = \pi$ is non-empty. We also define the set of couplings

$$\tilde{\Pi} = \{ \pi \in \Pi(\rho_0, \rho_1) : \pi = (id, T_\pi)_\#(\rho_0) \text{ for some map } T_\pi \in \mathcal{T}_{\rho_0, \rho_1} \}.$$

accumulated potential over the evolving density path. The cost associated with a coupling $\pi \in \tilde{\Pi}$ is then defined as

$$c(\pi) = \inf_{\gamma \in \Gamma(\pi)} \mathcal{A}(\gamma).$$

Our primary objective is the following action-minimizing problem:

$$\inf_{\pi \in \tilde{\Pi}(\rho_0, \rho_1)} c(\pi) = \inf_{\pi \in \tilde{\Pi}(\rho_0, \rho_1)} \inf_{\gamma \in \Gamma(\pi)} \mathcal{A}(\gamma). \tag{11}$$

To the best of our knowledge, this formulation is novel. While related ideas appear in [29] and [32], those works are limited to linear potentials or "lifted" Lagrangians. In contrast, our formulation includes internal and interaction energy terms, which significantly alter the structure of the problem. In particular, the cost of each particle trajectory depends not only on its velocity and position, but also on the global density path $\rho_t$.

The following theorem, whose proof can be found in Appendix B, establishes the equivalence between the dynamic and static formulations given in Equations (6) and (11), respectively.

**Theorem 1.** *The dynamic formulation in Equation* (6) *is equivalent to the static formulation in Equation* (11).

With these definitions in place, we now formulate the action-minimizing problem from a parametric perspective. Let the parameter space $\Theta$, the parametric map $T$, and the reference density $\lambda$ be fixed. Define the set of admissible boundary parameters as

$$\Theta_0^1 = \left\{ (\theta_0, \theta_1) \in \mathbb{R}^D \times \mathbb{R}^D : (T_{\theta_i})_\# \lambda = \rho_i, i = 0, 1 \right\}.$$

Given a parameter curve with boundary condition $\theta_{0 \to 1} \in \Theta_{0 \to 1} := \{ \theta_{0 \to 1} \in C^1([0,1], \Theta) : \theta(0) = \theta_0, \theta(1) = \theta_1 \}$, we define its action (by slight abuse of notation) as

$$\mathcal{A}(\theta_{0 \to 1}) := \mathbb{E}_{z \sim \lambda} \left[ \int_0^1 \left\| \frac{d}{dt} T_{\theta(t)}(z) \right\|^2 dt \right] + \int_0^1 F((T_{\theta(t)})_\# \lambda) dt, \tag{12}$$

where $T_{\theta(t)}(z)$ denotes the transported trajectory at $t$ of a particle $z$ drawn from the reference distribution with density $\lambda$. The corresponding parametric action-minimization problem is given by:

$$\inf_{(\theta_0, \theta_1) \in \Theta_0^1} \inf_{\theta_{0 \to 1} \in \Theta_{0 \to 1}} \mathcal{A}(\theta_{0 \to 1}). \tag{13}$$

This formulation parallels the coupling-based problem in Equation (11) but operates directly in the parameter space of the transport maps.

While [16] explored the Lagrangian formulation of Equation (6) in parameter space from a theoretical perspective, our approach in Equation (13) introduces an alternative formulation based on coupling costs. This perspective not only offers conceptual clarity but also lends itself naturally to numerical implementation.

Motivated by [20], we employ cubic Hermite spline approximation of the parameter curve $\theta_{0 \to 1}$ in solving the optimization problem (13). The following theorem provides an estimate for the error introduced in the objective action. The full set of assumptions and proof are provided in Appendix C.

**Theorem 2.** *Let $\theta_{0 \to 1}$ be a parameter curve with $l \geq 1$ continuous derivatives, and let $\tilde{\theta}_{0 \to 1}$ be its cubic Hermite spline approximation using $K$ uniformly spaced control (collocation) points. Then the error in the action satisfies $|\mathcal{A}[\theta_{0 \to 1}] - \mathcal{A}[\tilde{\theta}_{0 \to 1}]| = O(h^{\kappa - 1})$, where $\kappa := \min(l, 4)$.*

While action-minimizing curves can be computed analytically or via numerical integration for simple parametric families (e.g., affine maps) and linear potentials (see [33]), such formulations are inherently limited in expressivity. To overcome this, we adopt Neural ODEs [9] as flexible and expressive parametric maps for modeling complex density dynamics.

### 3.2 Pushforward map via Neural ODE

Neural Ordinary Differential Equations (Neural ODEs) [9] define continuous-time transformations via parameterized differential equations:

$$\frac{d\psi(\tau, z)}{d\tau} = v_\theta(\tau, \psi(\tau, z)), \quad \psi(0, z) = z \sim \lambda, \tag{14}$$

where $v_\theta : [0, 1] \times \mathbb{R}^d \to \mathbb{R}^d$ is a neural network parameterized by $\theta$, $\lambda$ is typically a standard normal distribution, and $\tau$ denotes the ODE time variable. The resulting transport map is defined as:

$$T_\theta(z) = z + \int_0^1 v_\theta(\tau, \psi(\tau, z)) \, d\tau. \tag{15}$$

We adopt Neural ODEs as parametric maps due to their ability to support efficient computation of both the entropy $\log(\rho)$ and the score $\nabla_x \log(\rho)$ through auxiliary ODE systems [35]. Details of these complementary entropy and score equations are provided in Appendix D.

### 3.3 Optimization framework

The static formulation (13) defines a bilevel optimization problem: the boundary conditions ensure feasibility by satisfying $(T_{\theta_i})_\# \lambda = \rho_i, i = 0, 1$. The curve $\theta_{0 \to 1}$ minimizes the action and determines the coupling cost. From the perspective of the spline control points $\{\theta_{t_i}\}_{i=0}^{K+1}$, the boundary parameters are $\theta_0, \theta_1$, while the interior path is defined by $\{\theta_{t_i}\}_{i=1}^{K}$. This motivates our optimization strategy:

**Step 1: Initialization**. First, initialize the boundary parameters $\theta_0$ and $\theta_1$ such that $(T_{\theta_i})_\# \lambda$ are good approximations of $\rho_i$ for $i = 0, 1$. We employ Flow Matching (FM) for its efficiency. We in general consider this step outside of the main algorithm. In Table 1 we include in the last column the FM training times. A parameter can be used for any functional, so once a boundary parameter has been obtained, it can be reused across multiple problems, see Appendix F.1.2. Second, initialize the control points $\{\theta_{t_i}\}_{i=1}^{K}$ as the linear interpolation of $\theta_0$ and $\theta_1$. Then run a few iterations of path optimization with $F(\rho) = 0$. In Appendix F.1.1 and Appendix F.2, we show the computational advantages of each part of the initialization step.

**Step 2: Path optimization**. Perform a gradient-based path optimization to update the interior control points $\{\theta_{t_i}\}_{i=1}^{K}$ to minimize the action (12), while keeping the boundary parameters fixed, see details

in Algorithm 1. The action functional (12) is evaluated using the trapezoidal rule with $N$ time steps and $M$ Monte Carlo samples. The gradient is computed using automatic differentiation. See Algorithm 1

**Step 3: Boundary/coupling optimization**. Optimize the boundary parameters $(\theta_0, \theta_1)$ while keeping the interior spline control points $\{\theta_{t_i}\}_{i=1}^{K}$ fixed, see details in Algorithm 2. Specifically, we formulate this as a penalized optimization problem:

$$\min_{\theta_0, \theta_1 \in \Theta} \mathbb{E}_{z \sim \lambda} \left[ \sum_{i=0}^{1} \alpha_i L((T_{\theta_i})_\# \lambda, \rho_i) \right] + \mathcal{A}[\theta_{0 \to 1}]. \tag{16}$$

Here $L$ is a dissimilarity metric between distributions, for which we use the FM loss function. The weights $\alpha_i$, $i = 0, 1$, balance boundary accuracy and coupling optimality. The action $\mathcal{A}[\theta_{0 \to 1}]$ (12) is added as a penalty term, which is estimated at the current control points, see Algorithm 2

We repeat the optimizations of step 2 and step 3 alternatively until convergence, producing both an optimal coupling and a minimal-action path between $\rho_0$ and $\rho_1$, see Algorithm 3. Observe as long as the action is lower bounded, step 2 will converge to a local minimum. The dissimilarity is assumed to be nonnegative, thus step 3 also converges to a local minimum.

# 4 Experiments

We conduct a comprehensive evaluation of our method across a variety of benchmark scenarios to demonstrate its accuracy, efficiency, flexibility, and scalability. In all experiments, we compare against GSBM [20], the current state-of-the-art, which has been shown to outperform earlier approaches. Where appropriate, we also include comparisons with Neural Lagrangian Optimal Transport (NLOT) [29] and APAC-Net [17]. It is important to note that NLOT is limited to linear potentials, while APAC-Net is specifically designed for mean-field games (MFG) and employs soft terminal costs instead of enforcing hard distributional constraints.

In F.3 we include an ablation study on the number of control points.

## 4.1 Implementation details

We implement our method in PyTorch [28], and run all experiments on an AMD 7543 CPU + NVIDIA RTX A6000 GPU. The Wasserstein-2 distance is approximated using the POT library [12] with 3,000 samples. For estimating the action, we use Monte Carlo integration with 3,000 samples and compute the time integral using the trapezoidal rule with a step size of $\Delta t = 1/50$. Each experiment is repeated across three random trials with different seeds. Additional implementation and experimental details are provided in Appendix F. In Table 4 see the boundary conditions $\rho_0$ and $\rho_1$, and Table 4 the hyper-parameters for the PDPO algorithm.

## 4.2 Obstacle avoidance with mean-field interactions

We demonstrate the effectiveness of our method through three challenging obstacle avoidance scenarios involving mean-field interactions. See Appendix F.4 for their mathematical definitions.

**S-Curve with Congestion (SCC)** from [17]: Particles navigate around two obstacles arranged in an "S"-shaped configuration while interacting with one another. PDPO achieves superior performance compared to existing methods (GSBM and APAC-Net), yielding lower action values, more accurate boundary approximation, and faster runtimes (see Table 1).

Notably, the interpolated density path generated by PDPO (Figure 2b) exhibits clear nonlinearity between control points. As shown in Figure 2a, a linear interpolation of the density between control points $\theta_{t_1}$ (red) and $\theta_{t_2}$ (orange) would intersect the obstacles. This highlights how PDPO's spline-based parameterization effectively exploits the geometry of the parameter space to avoid infeasible paths. Note that both GSBM and APAC-Net violate the obstacle avoidance constraint in this scenario.

**V-Neck with Entropy and Fisher Information (VNEFI)** from [20]: This stochastic optimal control (SOC) problem involves particles navigating a narrow channel while minimizing entropy, see Figure 11b. PDPO outperforms GSBM by achieving lower action values, comparable boundary accuracy, and significantly faster runtime (see Table 1). When GSBM is restricted to the same runtime as

Table 1: Comparison for obstacle avoidance with mean-field interactions. The quantities $W_2(\rho_i)$ $i = 0, 1$ denote the empirical are the Wasserstein-2 distances at times computed using the POT library [12]. The FM time is the sum of the FM training times for both boundaries.

| Problem (method) | $\mathcal{A}[\rho_t]$ | $\int_0^1 \mathbb{E}_{\rho_t} V(X)dt$ | $(W_2(\rho_0), W_2(\rho_1))$ | Time | FM Time |
|---|---|---|---|---|---|
| SCC (PDPO) | $\mathbf{39.90 \pm 0.16}$ | $1.12 \pm 0.02$ | $(\mathbf{0.028}, \mathbf{0.014})$ | $\mathbf{11m}$ | $134s$ |
| SCC (GSBM) | $40.02 \pm .75$ | $4.05 \pm 1.48$ | $(0.21, 0.25)$ | $21m$ | - |
| SCC (Apac-net) | $39.95 \pm 0.25$ | $1.31 \pm 0.16$ | $(0, 0.093)$ | $12m$ | - |
| VNEFI (PDPO) | $\mathbf{148.91 \pm 0.72}$ | $38.290 \pm 0.43$ | $(\mathbf{0.082}, 0.078)$ | $\mathbf{70m}$ | $77s$ |
| VNEFI (GSBM) | $156.80 \pm 4.31$ | $74.28 \pm 1.23$ | $(0.086, \mathbf{0.062})$ | $115m$ | - |
| GMM (PDPO) | $\mathbf{79.46 \pm 0.62}$ | $0.41 \pm 0.35$ | $(0, 0.359)$ | $\mathbf{8m}$ | $3m10s$ |
| GMM (GSBM) | $88.77 \pm 3.31$ | $14.59 \pm 3.26$ | $(0.806, \mathbf{0.258})$ | $123m$ | - |
| GMM (NLOT) | $82.06 \pm 2.80$ | $0.006 \pm 0.006$ | $(0, 0.629)$ | $117m$ | - |

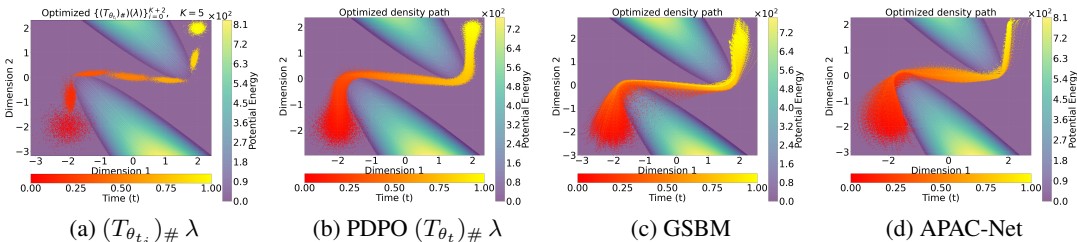

(a) $(T_{\theta_{t_i}})_{\#}\lambda$     (b) PDPO $(T_{\theta_t})_{\#}\lambda$     (c) GSBM     (d) APAC-Net

Figure 2: S-curve-C: (a) Pushforward densities at control points, (b) Pushforward densities along the interpolated trajectory, (c) Density path produced by GSBM, (d) Density path produced by APAC-Net

PDPO (details in Appendix F.4), its action value remains similar, but its boundary approximation deteriorates by an order of magnitude.

**Gaussian Mixture obstacle (GMM)** from [20]: In this benchmark, particles must move between multimodal source and target distributions while avoiding a Gaussian mixture-shaped obstacle, see Figure 13 for the density path in the appendix. This example demonstrates PDPO's flexibility in selecting different reference distributions. By setting the reference distribution $\lambda = \rho_0$, there is no approximation error at time $t = 0$. In Table 1 we compare against GSBM and NLOT. Our approach achieves a slight improvement of $0.04\%$ in the action, but the computational time was reduced by approximately $85\%$ from nearly 2 hours to less than 10 minutes. While GSBM achieves a marginally better boundary representation for $\rho_1$, our method offers comparable optimization quality with substantially faster computation.

Note that for GSBM, the reported $W_2(\rho_0)$ values are computed by evolving samples drawn from $\rho_1$ via the backward SDE to obtain the empirical approximation of $\rho_0$

### 4.3 Generalized momentum minimization

Building on the momentum Schrödinger Bridge (SB) framework by [10], we extend action-minimizing problems to incorporate acceleration alongside additional potential terms. This leads to a novel formulation that minimizes the following objective:

$$\mathcal{A}(\gamma) = \mathbb{E}_{x \sim \rho_0} \left[ \int_0^1 \left\| \frac{d^2}{dt^2} \gamma_t(x) \right\|^2 dt \right] + \int_0^1 F(\rho_{\gamma_t})dt.$$

Figure 3 illustrates PDPO solutions for the obstacle avoidance with mean-field interaction class of problems. The resulting paths are noticeably smoother, compare the top and bottom rows of Figure 3

To our knowledge, no prior method has addressed these generalized settings within a unified momentum formulation, underscoring PDPO's flexibility beyond classical Wasserstein geodesics.

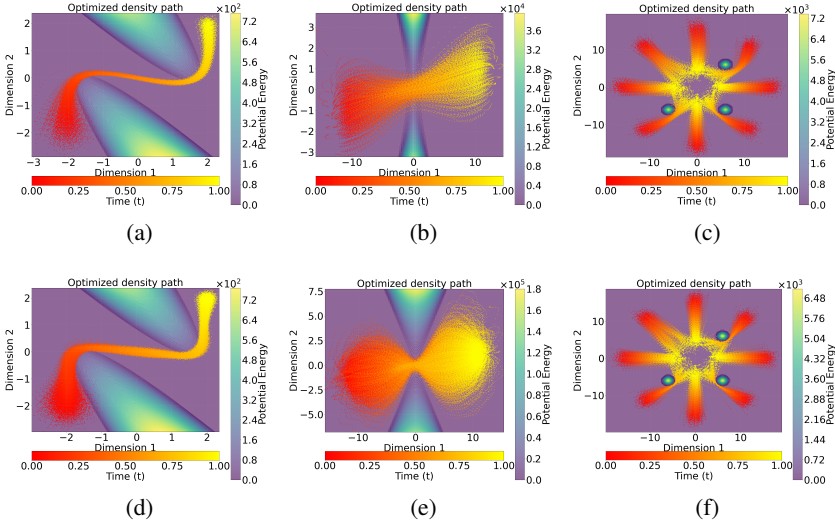

Figure 3: Solutions for generalized momentum-minimizing problems (a-c). Solutions for kinetic energy minimizing problems (d-f).

## 4.4 High-dimensional opinion depolarization

We examine the opinion depolarization problem following [30] and [19]. Here, $\rho(t, \cdot) : \mathbb{R}^d \to \mathbb{R}_{\geq 0}$ models the opinion density of a population, where each coordinate represents an agent's opinion on a topic. The particle-level opinion dynamics evolve as

$$\frac{dx(t)}{dt} = \frac{f_{\text{polarize}}(x(t); \rho(t, \cdot))}{\|f_{\text{polarize}}(x(t); \rho(t, \cdot))\|},$$

with

$$f_{\text{polarize}}(x; \rho(t, \cdot), \xi_t) := \mathbb{E}_{y \sim p_t}[a(x, y, \xi_t)\bar{y}], \quad a(x, y, \xi_t) := \begin{cases} 1 & \text{if } \text{sign}(\langle x, \xi_t \rangle) = \text{sign}(\langle y, \xi_t \rangle) \\ -1 & \text{otherwise} \end{cases}.$$

The function $a(x, y; \xi)$ evaluates opinion alignment at random information $\xi$ sampled independently of $\rho(t, \cdot)$. These dynamics cause an unpolarized initial condition $\rho(0, \cdot)$ (e.g., Gaussian) to polarize into opinion clusters. We seek a velocity $v(t, x)$ preventing polarization by enforcing an unpolarized terminal condition $\rho(1, \cdot)$ (see Appendix F.5).

The action is

$$\mathcal{A}_{\text{polarize}}(\theta_{0 \to 1}) := \mathbb{E}_{z \sim \lambda} \left[ \frac{1}{2} \|f_{\text{polarize}}(T_{\theta(t)}(z); (T_{\theta(t)})_\# (\lambda)) - \frac{d}{dt} T_{\theta(t)}(z)\|^2 \right] + \int_0^1 F((T_{\theta(t)})_\# (\lambda)) dt.$$

Figure 4 shows PDPO achieves quality comparable to [20], as evidenced by 2D PCA projections and directional similarity histograms at terminal time. These histograms show cosine similarity distributions between opinion vector pairs, measuring alignment independent of magnitude. PDPO's key advantage is computational efficiency: completing in 23 minutes versus over 5 hours for GSBM. We attribute this success to: (1) the spline path $\theta_{0 \to 1}$ readily reaches the non-polarized target at $t = 1$ from initialization, and (2) it connects boundary conditions independently of $f_{\text{polarize}}$. Note that training boundary conditions $\theta_0$ and $\theta_1$ required 29m35s and 31m12s, respectively.

## 5 Computational complexity and limitations

The primary computational cost in our algorithm arises from pushforward evaluations. When the action integral is approximated using $N$ time points, our method requires evaluating $N$ Neural ODEs (NODEs). Each NODE involves integrating a velocity field using 10 time steps. For expectation estimates, we use $M$ Monte Carlo samples per NODE.

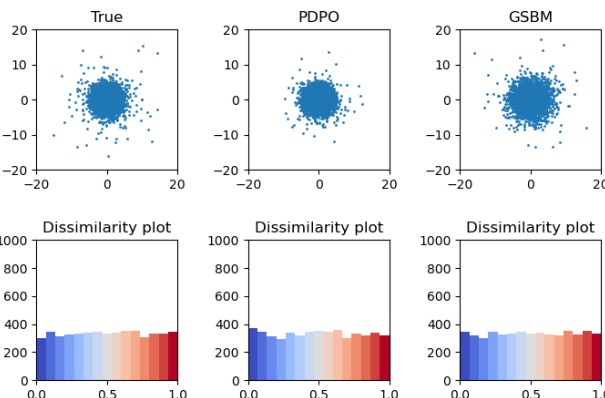

Figure 4: Opinion depolarization in 1000 dimensions. Top: 2D PCA projections of the distributions. Bottom: directional similarity histograms. Left: target unimodal distribution. Middle: PDPO solution. Right: DeepGSB solution.

When the functional $F$ includes entropy and Fisher Information terms, the cost increases substantially. Computing the entropy requires solving an additional 1D ODE, while the Fisher Information involves a separate $d$-dimensional ODE, where $d$ is the problem dimension. In total, this results in $O(N \times M \times (1 + d))$ ODE evaluations for problems involving both terms.

In Appendix F.6 , we showcase the scalability of our method with a 50-dimensional Schrödinger Bridge example, representing the highest-dimensional case with Fisher Information we have successfully computed.

Currently, our implementation is limited to NODEs with MLP architectures, see Appendix F, which limits applications to image-based transport problems, unlike GSBM [20].

# 6 Conclusions

We introduced Parametric Density Path Optimization (PDPO), a general framework for solving action-minimizing problems over probability densities by leveraging parameterized pushforward maps. This approach transforms the original infinite-dimensional optimization over density paths into a tractable, finite-dimensional problem in parameter space, using cubic splines with a small number of control points.

A central conceptual distinction of PDPO is that its outcome is not a single trained neural network, but a collection of $K + 1$ trained models $\{T_{\theta_{t_i}}\}_{i=0}^{K}$ whose parameters lie on a spline-defined trajectory. Together, these models define a continuous transformation in density space through the pushforward of the interpolated parameters. From this perspective, PDPO learns a dynamical path in the parameter space, a time-dependent family of mappings that approximate the evolution of the probability mass.

Experimental results show that PDPO matches or surpasses state-of-the-art methods in accuracy, while requiring substantially less computation time.

# 7 Broader Impact

The study of action-minimizing problems in density space might have applications in the study of population dynamics.

## Acknowledgments and Disclosure of Funding

Chen acknowledges partial support from NSF #2325631, #2245111. This research is partially supported by NSF grant DMS-230746. The authors would also like to acknowledge Benjamin Burns for the helpful discussions and the rendering of Figure 1a and Shu Liu for the helpful discussions.

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

## A  Deterministic and SOC problems

We now show the equivalence of

$$\inf_{\rho,v} \int_0^1 \int_{\mathbb{R}^d} \frac{1}{2}\|v(t,x)\|^2 \rho(t,x)dx + F(\rho(t)) + \mathcal{FI}(\rho(t))\, dt \tag{17}$$
$$\text{subject to} \quad \partial_t \rho + \nabla \cdot (\rho v) = 0, \quad \rho(0,\cdot) = \rho_0, \quad \rho(1,\cdot) = \rho_1,$$

and

$$\inf_{\rho(t,x),u(t,x)} \int_0^1 \int_{\mathbb{R}^d} \frac{1}{2}\|u(t,x)\|^2 \rho(t,x)dx + F(\rho(t))\, dt \tag{18}$$
$$\text{subject to} \quad \partial_t \rho + \nabla \cdot (\rho u) = \frac{\sigma^2}{2}\Delta\rho, \quad \rho(0,\cdot) = \rho_0, \quad \rho(1,\cdot) = \rho_1.$$

*Proof.* In what follows, we use the notation $\delta_\rho$ for the $L^2$ gradient.

Let $S$ be a Lagrange multiplier for (17), then KKT conditions for (17) are given by

$$v = \nabla S \quad \rho, \text{ a.e.}$$
$$\partial_t S + \frac{1}{2}\|\nabla S\|^2 = -\delta_\rho F(\rho) - \delta_\rho \mathcal{FI}(\rho),$$
$$\partial_t \rho + \nabla \cdot (\rho v) = 0.$$

Define $\Phi = S + \sigma^2/2 \log \rho$, and $u := \nabla\Phi$, $\rho$ a.e. Then the pair $(\rho, u)$ satisfies

$$u = \nabla\Phi, \quad \rho \text{ a.e.}$$
$$\partial_t \Phi + \frac{1}{2}\|\nabla\Phi\|^2 = -\delta_\rho F(\rho),$$
$$\partial_t \rho + \nabla \cdot (\rho u) = \frac{\sigma^2}{2}\Delta\rho,$$

which is in turn the KKT conditions of the optimization problem (18). $\qquad\square$

## B  Equivalence of dynamic and static frameworks

Here we show the equivalence between

$$\inf_{\rho,v} \int_0^1 \int_{\mathbb{R}^d} \frac{1}{2}\|v(t,x)\|^2 \rho(t,x)dx + F(\rho(t))\, dt$$
$$\text{subject to} \quad \partial_t \rho + \nabla \cdot (\rho v) = 0, \quad \rho(0,\cdot) = \rho_0, \quad \rho(1,\cdot) = \rho_1,$$

and

$$\inf_{\pi \in \tilde{\Pi}(\rho_0,\rho_1)} c(\pi) = \inf_{\pi \in \tilde{\Pi}(\rho_0,\rho_1)} \inf_{\gamma \in \Gamma(\pi)} \mathcal{A}(\gamma).$$

*Proof.* Observe that a local minimizer of (6) $(\rho, v)$ defines a $C^1$-diffeomorphic curve $\gamma_{\rho,v}$ as the flow

$$\gamma(t,x) = x + \int_0^t v(s,\gamma(s,x))ds,$$

and the coupling $\pi_{\rho,v} = (x, \gamma(1,x))$. The coupling $\pi_{\rho,v}$ and path $\gamma_{\rho,v}$ are an admissible pair for (11). From the definition of $\gamma_{\rho,v}$, it follows that $\rho_{\gamma_{\rho,v}(t)} = (\gamma_{\rho,v}(t))_\#(\rho_0) = \rho$, and

$$\int_0^1 \int_{\mathbb{R}^d} \frac{1}{2}\|v(t,x)\|^2 \rho(t,x)dxdt = \mathcal{A}(\gamma_{\rho,v}),$$

then

$$\int_0^1 \int_{\mathbb{R}^d} \frac{1}{2}\|v(t,x)\|^2 \rho(t,x)dx + F(\rho)\, dt = \mathcal{A}(\gamma_{\rho,v}) + \int_0^1 F(\rho_{\gamma,v})dt.$$

Thus (6) upper bounds (11). Now, given a local minimizer of (11) $(\pi, \gamma)$, the density $\rho_{\pi,\gamma}(t, \cdot) = (\gamma_t(\cdot))_\#(\rho_0)$ and the velocity field

$$v(t, \gamma(t,x)) = \frac{d}{dt}\gamma(t,x), \quad \gamma(0,x) = x \sim \rho_0,$$

define an admissible pair of (6). As before,

$$\mathcal{A}(\gamma) + \int_0^1 F(\rho_\gamma)dt = \int_0^1 \int_{\mathbb{R}^d} \frac{1}{2}\|v_{\pi,\gamma}(t,x)\|^2 \rho_{\pi,\gamma}(t,x)dx + F(\rho_{\pi,\gamma})dt.$$

Thus, (11) upper bounds (6). $\qquad\square$

## C   Wasserstein error bounds for interpolation

**Theorem 3.** *Assume that $T$ is Lipschitz in its second variable with a constant $C$, in the sense, $\forall z \in \mathbb{R}^d$ and any $\theta, \tilde{\theta} \in \mathbb{R}^D$,*

$$\|T(z,\theta) - T(z,\tilde{\theta})\| \leq C\|\theta - \tilde{\theta}\|,$$

*then*

$$W_2(\rho_\theta, \rho_{\tilde{\theta}}) \leq C\|\theta - \tilde{\theta}\|. \tag{19}$$

*Proof.* From the static definition of the Wasserstein distance Equation (3), we have

$$W_2(\rho_\theta, \rho_{\tilde{\theta}}) \leq \left(\int_{\mathbb{R}^d} \|T(z,\theta) - T(z,\tilde{\theta})\|^2 \lambda(z)dz\right)^{1/2} \leq C\|\theta - \tilde{\theta}\|.$$

$\qquad\square$

We recognize that $T$ being Lipschitz in the above sense is strong. We leave for future research, relaxing it to locally Lipschitz, a more feasible condition.

**Corollary 1.** *Let $\theta_{0\to1}$ and $\tilde{\theta}_{0\to1}$ denote two curves in the parameter space. Following the hypothesis on Theorem 3, it follows that $\forall t \in [0,1]$*

$$W_2(\rho_{\theta(t)}, \rho_{\tilde{\theta}(t)}) \leq C\|\theta(t) - \tilde{\theta}(t)\|.$$

**Corollary 2.** *Let $\theta_{0\to1}$ be a curve in parameter space with continuous derivatives up to order $l \geq 1$, and let $\tilde{\theta}_{0\to1}$ be the piecewise cubic Hermite spline interpolation of $\theta(t)$ at $K$ uniformly spaced points $\{t_i = \frac{i}{K+1}\}_{i=0}^{K+1}$. Define $\kappa := \min(l, 4)$, then*

$$\|\theta_{0\to1} - \tilde{\theta}_{0\to1}\| \leq M_\kappa \max_{\xi \in [0,1]} \|\theta^{(\kappa)}(\xi)\| h^\kappa,$$

*where $h = \frac{1}{K+1}$ and $M_\kappa$ is a constant depending on $\kappa$. With $\theta^{(\kappa)}$, the $\kappa$ derivative of the path $\theta(t)$. Furthermore for all $t \in [0,1]$, the Wasserstein distance between the density paths at time $t$ is given by*

$$W_2(\rho_{\theta(t)}, \rho_{\tilde{\theta}(t)}) \leq CM_\kappa \max_{\xi \in [0,1]} \|\theta^{(\kappa)}(\xi)\| h^\kappa,$$

*where $C$ is the Lipschitz constant from Theorem 3.*

**Theorem 4.** *Let $\theta_{0\to1}$ be a curve in parameter space with continuous derivatives up to order $l \geq 2$, and let $\tilde{\theta}_{0\to1}$ be the cubic Hermite spline interpolation of $\theta(t)$ at $K$ uniformly spaced points $\{t_i = \frac{i}{K+1}\}_{i=0}^{K+1}$. Define $\kappa := \min(l, 4)$.*

*Assume that:*

1. *$F$ is Lipschitz with respect to the Wasserstein-2 distance with constant $M_F$.*

2. *$\|\partial_\theta T(z,\theta)\| < M_{\partial_\theta T}$, for some constant $M_{\partial_\theta T}$ for all $z$ and $\theta$.*

3. *$\partial_\theta T(z,\theta)$ is Lipschitz continuous in $\theta$ for all $z$ with Lipschitz constant $C_{\partial_\theta}$.*

4. $\|\theta'(t)\| < M_{\partial_{\theta'}}$, for some constant $M_{\partial_{\theta'}}$, for all $t \in [0, 1]$.

*Then the error in the action functional is bounded by:*

$$|A[\theta_{0\to1}] - A[\tilde{\theta}_{0\to1}]| \leq C_{\mathcal{A}} \max_{\xi\in[0,1]} \|\theta^{(\kappa)}(\xi)\| h^{\kappa-1},$$

*where $h = \frac{1}{K+1}$ and $C_{\mathcal{A}}$ is defined below at the end of the proof.*

*Proof.* We split the proof into two parts, one part on a bound for the kinetic energy term, and the other on a bound for the potential energy term. The bound for the potential energy follows from the Lipschitz continuity of $F$ and Corollary 1

$$\int_0^1 |F(\rho_{\theta(t)}) - F(\rho_{\tilde{\theta}(t)})|dt \leq \int_0^1 M_F W_2(\rho_{\theta(t)}, \rho_{\tilde{\theta}(t)})dt \leq M_F M_\kappa \cdot h^\kappa \max_{\xi\in[0,1]} \|\theta^{(\kappa)}(\xi)\|.$$

The bound for the kinetic energy can be derived by assumptions 2–4 as

$$\int_0^1 \left| \mathbb{E}_{z\sim\lambda} \left\| \frac{d}{dt} x_{\theta(t)}(z) \right\|^2 - \mathbb{E}_{z\sim\lambda} \left\| \frac{d}{dt} x_{\tilde{\theta}(t)}(z) \right\|^2 \right| dt$$

$$\leq \int_0^1 \mathbb{E}_{z\sim\lambda} \left| \left\| \frac{d}{dt} x_{\theta(t)}(z) \right\|^2 - \left\| \frac{d}{dt} x_{\tilde{\theta}(t)}(z) \right\|^2 \right| dt$$

$$\leq \int_0^1 \mathbb{E}_{z\sim\lambda} \left[ \left( \left\| \frac{d}{dt} x_{\theta(t)}(z) \right\| + \left\| \frac{d}{dt} x_{\tilde{\theta}(t)}(z) \right\| \right) \left\| \frac{d}{dt} x_{\theta(t)}(z) - \frac{d}{dt} x_{\tilde{\theta}(t)}(z) \right\| \right] dt$$

$$\leq \int_0^1 \mathbb{E}_{z\sim\lambda} \Big[ \Big( \|(\partial_\theta T)(z, \theta(t))\| \|\theta'(t)\| + \|(\partial_\theta T)(z, \tilde{\theta}(t))\| \|\tilde{\theta}'(t)\| \Big)$$

$$\cdot \|(\partial_\theta T)(z, \theta(t))\theta'(t) - (\partial_\theta T)(z, \tilde{\theta}(t))\tilde{\theta}'(t)\| \Big] dt$$

$$\leq (M_{\partial_\theta T} M_{\theta'} + M_{\partial_\theta T} M_{\tilde{\theta}'}) \int_0^1 \mathbb{E}_{z\sim\lambda} \|(\partial_\theta T)(z, \theta(t))\theta'(t) - (\partial_\theta T)(z, \tilde{\theta}(t))\tilde{\theta}'(t)\| dt$$

$$= M_{\partial_\theta T}(M_{\theta'} + M_{\tilde{\theta}'}) \int_0^1 \mathbb{E}_{z\sim\lambda} \|(\partial_\theta T)(z, \theta(t))\theta'(t) - (\partial_\theta T)(z, \tilde{\theta}(t))\tilde{\theta}'(t)\| dt$$

$$\leq M_{\partial_\theta T}(M_{\theta'} + M_{\tilde{\theta}'}) \int_0^1 \mathbb{E}_{z\sim\lambda} \Big[ \|(\partial_\theta T)(z, \theta(t))\theta'(t) - (\partial_\theta T)(z, \theta(t))\tilde{\theta}'(t)\|$$

$$+ \|(\partial_\theta T)(z, \theta(t))\tilde{\theta}'(t) - (\partial_\theta T)(z, \tilde{\theta}(t))\tilde{\theta}'(t)\| \Big] dt$$

$$\leq M_{\partial_\theta T}(M_{\theta'} + M_{\tilde{\theta}'}) \int_0^1 \mathbb{E}_{z\sim\lambda} \Big[ \|(\partial_\theta T)(z, \theta(t))\| \|\theta'(t) - \tilde{\theta}'(t)\|$$

$$+ \|(\partial_\theta T)(z, \theta(t)) - (\partial_\theta T)(z, \tilde{\theta}(t))\| \|\tilde{\theta}'(t)\| \Big] dt$$

$$\leq M_{\partial_\theta T}(M_{\theta'} + M_{\tilde{\theta}'}) \int_0^1 \Big( M_{\partial_\theta T} \|\theta'(t) - \tilde{\theta}'(t)\| + C_{\partial_\theta} \|\theta(t) - \tilde{\theta}(t)\| \|\tilde{\theta}'(t)\| \Big) dt$$

$$\leq M_{\partial_\theta T}(M_{\theta'} + M_{\tilde{\theta}'}) \left( M_{\partial_\theta T} M_{\kappa-1} h^{\kappa-1} + C_{\partial_\theta} M_\kappa h^\kappa M_{\tilde{\theta}'} \right) \max_{\xi\in[0,1]} \|\theta^{(\kappa)}(\xi)\|.$$

A combination of the above two bounds concludes with

$$C_{\mathcal{A}} = \max\{M_F M_\kappa h, M_{\partial_\theta T}(M_{\theta'} + M_{\tilde{\theta}'})(M_{\partial_\theta T} M_{\kappa-1} + C_{\partial_\theta} M_\kappa M_{\tilde{\theta}'} h)\}.$$

$\square$

We recognize that hypothesis 1 on Theorem 4, which requires $F$ to be Lipschitz with respect to the $W_2$ metric, is a strong assumption for general functionals $F$. When $F$ consists of linear potentials and bilinear interaction potentials whose corresponding $V(x)$ and $W(x-y)$ are Lipschitz continuous with constants $M_V$ and $M_W$, the functional $F(\rho) = \int_{\mathbb{R}^d} V(x)\rho(x)dx + \int_{\mathbb{R}^d\times\mathbb{R}^d} W(x-y)\rho(x)\rho(y)dxdy$

is also Lipschitz. To show this, let $\mu, \nu \in \mathcal{P}(\mathbb{R}^d)$ and $T_{\mu \to \nu}$ denote the Monge map between them, then

$$|F(\mu) - F(\nu)| =$$
$$= |\mathbb{E}_{X \sim \mu}[V(X)] + \mathbb{E}_{X \sim \mu}\mathbb{E}_{Y \sim \mu}[W(X - Y)] - (\mathbb{E}_{X \sim \nu}[V(X)] + \mathbb{E}_{X \sim \nu}\mathbb{E}_{Y \sim \nu}[W(X - Y)])|$$
$$\leq \mathbb{E}_{X \sim \mu}|V(X) - V(T_{\mu \to \nu}(X))| + \mathbb{E}_{X \sim \mu}\mathbb{E}_{Y \sim \mu}|W(X - Y) - W(T_{\mu \to \nu}(X) - T_{\mu \to \nu}(Y))|$$
$$\leq (M_V + 2M_W)W_2(\mu, \nu).$$

Observe that the last inequality follows from using the Lipschitz condition on $V$ and $W$, and the following application of Hölder inequality,

$$\mathbb{E}_{X \sim \mu}[|X - T_{\mu \to \nu}(X)|] \leq (\mathbb{E}_{X \sim \mu}[|X - T_{\mu \to \nu}(X)|^2])^{1/2}(\mathbb{E}_{X \sim \mu}[1^2])^{1/2} = (\mathbb{E}_{X \sim \mu}[|X - T_{\mu \to \nu}(X)|^2])^{1/2}.$$

## D   NODEs

See [9] for the standard reference on NODEs. For the NODE

$$\frac{d\psi(\tau, z)}{d\tau} = v_\theta(\tau, \psi(\tau, z)), \quad \psi(0, z) = z \sim \lambda,$$

initialized at $\rho(0, z) \sim \mathcal{N}(0, I)$, let $\rho(\tau, \cdot) := \psi(\tau, \cdot)_{\#}\lambda$, then

$$\frac{d}{d\tau}\log(\rho(\tau, x))|_{x=\psi(\tau, z)} = -\nabla \cdot v_\theta(\tau, \psi(\tau, z))), \tag{20}$$

$$\text{and} \quad \frac{d(\nabla_x \log \rho(\tau, x))}{d\tau}\Big|_{x=\psi(\tau, z)} = -\nabla(\nabla \cdot v_\theta(\tau, \psi(\tau, z)))$$
$$- (\nabla v_\theta(\tau, \psi(\tau, z)))^T \nabla_x \log \rho(\tau, x)|_{x=\psi(\tau, z)} \tag{21}$$

with initial conditions $\log(\rho(0, z)) = \log(\rho_{\mathcal{N}}(z))$, where $\rho_{\mathcal{N}}(z)$ is the density of a standard normal distribution, and $\nabla_x \log \rho(0, x)|_{x=z} = -z$ respectively. Although it is not necessary to use a Gaussian density for the initial condition, it is common in the literature, these quantities might not be accessible for other densities.

The result for (20) is standard and we do not include the proof here. Although Equation (21) has been proved before, see e.g., [35], we provide a proof for completeness.

*Proof.* For this proof, we assume $\rho(\tau, x)$ has continuous second-order derivatives with respect to $\tau$ and $x$. In what follows, we use the short hand notation $\psi$ to indicate $\psi(\tau, z)$, and $v$ instead of $v_\theta$.

$$\frac{d}{d\tau}\nabla_x \log(\rho) = \partial_\tau \nabla_x \log(\rho) + \nabla_x^2 \log(\rho)\, v$$
$$= \nabla_x \rho^{-1} \partial_\tau \rho + \nabla_x^2 \log(\rho)\, v$$
$$= -\nabla_x \rho^{-1} \nabla \cdot (\rho v) + \nabla_x^2 \log(\rho)\, v$$
$$= -\nabla_x(\nabla_x \log(\rho)^T v + \nabla_x \cdot v) + \nabla_x^2 \log(\rho)\, v$$
$$= -\nabla_x v^T \log(\rho) - \nabla_x(\nabla_x \cdot v).$$

The first equality follows from the dynamical system $\frac{d\psi}{d\tau} = v(\tau, \psi(\tau, z))$, the third equality follows from the continuity equation $\partial_\tau \rho = -\nabla \cdot (\rho v)$, and the fourth equality from $\rho^{-1}\nabla_x \rho = \nabla_x \log(\rho)$. $\square$

## E   Algorithms/Implementation details

We use two separate optimizers: one for the boundary conditions $(\theta_0, \theta_1)$ (Algorithm 2) and another for the control points $\{\theta_{t_i}\}_{i=1}^K$ (Algorithm 1). We use Adam optimizers [26] in both cases. For the learning rate scheduler, we used StepLR or cosine, with specifics provided for each experiment.

A key implementation challenge is preserving PyTorch's computational graph. Specifically, the 'torch.load' operation does not retain the graph for gradient-based optimization. This is particularly

problematic in Algorithm 1, where the interpolated points $\theta(t_j)$ used in the trapezoidal integration must track the computational graph with the spline control points to enable proper gradient flow during optimization. When 'torch.load' is used to assign models, this computational graph is lost, disconnecting the density path from the control point optimization.

To address this issue, we re-implemented the MLP architecture from scratch, passing weights and biases as explicit function arguments rather than storing them as internal model parameters, see Algorithm 4. While effective, this approach introduces limitations to architectural flexibility: testing new network architectures requires either hard-coding them or developing alternative methods to preserve computational graphs when loading pre-trained models. We leave overcoming this limitation to future research.

---

**Algorithm 1** Path optimization

---

**Require:** Parametric points $\{\theta_{t_i}\}_{i=0}^{K+1}$, potential function $F$, $N$ samples$\{z_i\}_{i=1}^{M} \sim \lambda$, number of optimization steps $Q_1$.
1: Initialize spline in parameter space with $\{\theta_{t_i}\}_{i=0}^{K+1}$.
2: **for** number of iterations $Q_1$ **do**
3:     Obtain $\{\theta(t_j)\}_{j=0}^{N}$ by evaluating the current spline at time steps $t_j = \frac{j}{N}$.
4:     Evaluate the $N+1$ pushforwards $\{T_{\theta(t_j)}(z_i)\}_{i=0,j=1}^{N,M}$.
5:     **if** Entropy or Fisher Information **then**
6:         Obtain entropy Equation (20) score from (21).
7:     **end if**
8:     Evaluate the Equation (12) by integrating in time by trapezoidal rule and evaluating the expectations via Monte Carlo.
9:     Update weights $\{\theta_{t_i}\}_{i=1}^{K}$ via gradient descent.
10: **end for**
11: **return** $\{\theta_{t_i}\}_{i=1}^{K}$

---

**Algorithm 2** Coupling optimization

---

**Require:** Sampling function from $\rho_0$, sampling function from $\rho_1$, potential function $F$, control points $\{\theta_{t_i}\}_{i=0}^{K+1}$, weights $\alpha_0, \alpha_1$, samples$\{z_k\}_{k=1}^{M} \sim \lambda$, number of optimization steps $Q_2$.
1: **for** number of iterations $Q_2$ **do**
2:     Sample $M$ points from $\{x_0^k\}_{k=1}^{M} \sim \rho_0$, and $\{x_1^k\}_{k=1}^{M} \sim \rho_1$.
3:     Evaluate $\ell = \sum_{j=0}^{1} \frac{1}{M} \left[ \sum_{k=1}^{M} \alpha_j L((T_{\theta_j}(z_k), x_j^k) \right]$.
4:     Obtain $\{\theta(t_j)\}_{j=0}^{N}$ by evaluating the current spline at time steps $t_j = \frac{j}{N}$.
5:     Evaluate the $N+1$ pushforwards $\{T_{\theta(t_j)}(z_k)\}_{i=0,k=1}^{N,M}$.
6:     **if** Entropy or Fisher Information **then**
7:         Obtain entropy Equation (20) score from (21).
8:     **end if**
9:     Evaluate the Equation (12) by integrating in time by trapezoidal rule and evaluating the expectations via Monte Carlo.
10:     Update $\theta_0, \theta_1$ by minimizing (16) via gradient descent.
11: **end for**
12: **return** $\theta_0, \theta_1$

---

### E.1 Hyperparameter Sensitivity and Tuning

**Neural ODE architecture:** The neural ODE model defines the parameter space and must be sufficiently expressive to capture the boundary distributions. Key considerations:

- **Architecture sizing:** Use larger networks than minimally required for boundary fitting. For example, while a [2,64,4] architecture can learn our V-neck boundaries accurately, we needed [2,128,4] to capture the solution complexity when potential terms significantly alter the density shape.

---

**Algorithm 3** PDPO

---

**Require:** Sampling function for $\rho_0$, sampling function for $\rho_1$, reference density $\lambda$ potential function
 $F$, number of control points $K$, weights $\alpha_0, \alpha_1$, initialized $\theta_0, \theta_1$, number of total iterations $Q$,
 number of path optimization steps $Q_1$, number of coupling optimization steps $Q_2$, number of
 geodesic-warmup steps $Q_3$.

1: Initialize $\theta_{t_i}$ using an equispaced linear interpolation of $\theta_0$ and $\theta_1$.
2: Run $Q_3$ steps of geodesic warmup.
3: **for** number of iterations $Q$ **do**
4:     Sample $M$ points $\{z_i\}_{i=1}^M \sim \lambda$,
5:     Update $\{\theta_{t_i}\}_{i=1}^K$ by minimizing the action of the points $\{z_i\}_{i=1}^M$ using Algorithm 1.
6:     Update $(\theta_0, \theta_1)$ using the points $\{z_i\}_{i=1}^M$ in Algorithm 2.
7: **end for**
8: **return** $\{\theta_{t_i}\}_{i=0}^{K+1}$.

---

---

**Algorithm 4** Parameterized MLP Forward Pass

---

**Require:** Architecture arch $= [d, w, L]$ where $d$ is input/output dimension, $w$ is hidden width, $L$ is
 number of layers
**Require:** Input $x \in \mathbb{R}^{n \times d}$, parameter vector $\theta \in \mathbb{R}^p$
**Require:** Time-varying flag time_varying $\in \{\text{true}, \text{false}\}$

1: $d_{\text{in}} \leftarrow$ if time_varying $d + 1$ else $d$                                  ▷ Adjust input dimension
2: idx $\leftarrow 0$                                                                          ▷ Current position in $\theta$
3: $h \leftarrow x$

4: **// Input Layer**
5: Extract $W_1 \in \mathbb{R}^{w \times d_{\text{in}}}$ from $\theta[\text{idx} : \text{idx} + w \cdot d_{\text{in}}]$
6: idx $\leftarrow$ idx $+ w \cdot d_{\text{in}}$
7: Extract $b_1 \in \mathbb{R}^w$ from $\theta[\text{idx} : \text{idx} + w]$
8: idx $\leftarrow$ idx $+ w$
9: $h \leftarrow \sigma(W_1 h + b_1)$                                                           ▷ $\sigma$ is activation function

10: **// Hidden Layers**
11: **for** $\ell = 2$ **to** $L - 1$ **do**
12:     Extract $W_\ell \in \mathbb{R}^{w \times w}$ from $\theta[\text{idx} : \text{idx} + w^2]$
13:     idx $\leftarrow$ idx $+ w^2$
14:     Extract $b_\ell \in \mathbb{R}^w$ from $\theta[\text{idx} : \text{idx} + w]$
15:     idx $\leftarrow$ idx $+ w$
16:     $h \leftarrow \sigma(W_\ell h + b_\ell)$
17: **end for**

18: **// Output Layer**
19: Extract $W_L \in \mathbb{R}^{d \times w}$ from $\theta[\text{idx} : \text{idx} + d \cdot w]$
20: idx $\leftarrow$ idx $+ d \cdot w$
21: Extract $b_L \in \mathbb{R}^d$ from $\theta[\text{idx} : \text{idx} + d]$
22: $h \leftarrow W_L h + b_L$                                                                  ▷ No activation on output
23: **return** $h$

---

- **Rule of thumb:** If the potential energy terms are expected to create new modes or dramatically change boundary shapes, increase the hidden dimension.

**Neural ODE Integration:** Two factors affect forward simulation quality:

- **Solver choice:** We use midpoint integration, as it is standard in flow-based generative models.

- **Integration steps:** 10 steps suffice for most problems. Use at least the number of steps required by your boundary NODEs for accurate sampling from the target distributions.

**Action Approximation (Time Steps N):** This is an important parameter that controls the temporal discretization accuracy.

Table 2: Action convergence with respect to time discretization steps N.

| N | Action | Kinetic Energy | Potential Energy |
|----|--------|----------------|------------------|
| 10 | 66.06 | 37.38 | 28.68 |
| 20 | 42.09 | 37.77 | 12.82 |
| 30 | 30.31 | 29.46 | 0.788 |
| 40 | 30.20 | 29.72 | 0.735 |
| 50 | 30.10 | 29.37 | 0.725 |

**Recommendation:** Start with N=20 minimum and increase until the action converges (typically N=30-50). The dramatic drop from N=10 to N=20 demonstrates why sufficient discretization is crucial. This parameter is related to the complexity of the problem: longer distances and more complex energy landscapes require finer temporal discretization.

**Monte Carlo Samples M:** The number of samples M affects the quality of the action estimate. Even for high-dimensional problems (e.g., 50D with M=1000), we maintain solution quality as shown in the Appendix.

**Computation of** $\log(\rho)$ **and** $\nabla \log(\rho)$**:** Following the scaling approach in the CNF literature [4], we use the Hutchinson trace estimator to make the ODE for $\log(\rho)$ computationally tractable. To the best of our knowledge, there are no unbiased estimator techniques for $\nabla \log(\rho)$, and we consider developing such estimators an interesting direction for future work.

# F  Additional Numerical Results

In this subsection, we report all the experimental details. In Table 4 we report the hyperparameters for the algorithms and the boundary conditions for each problem. The notation [x,y,z] in Table 4 defines the architecture of the networks, x is the input dimension, y is the number of neurons per layer, and z is the number of layers. We assumed the value of the constants $\alpha_0 = \alpha_1$ in Algorithm 2, which we report as $\alpha$ in Table 4.

Here we report that [20] is under CC BY-NC licence, [29] is under CC BY-NC 4.0 License, and [17] has no license.

## F.1  Pre-training strategy

### F.1.1  Zero initialized boundary parameters

Our framework can use pre-trained parameters $\theta_0$ and $\theta_1$ for the boundary conditions $\rho_0$ and $\rho_1$, respectively. To pre-train a parameter, we use Flow Matching (FM) [18] to learn a velocity field to sample from a dataset. The choice of training framework for the boundary parameters motivates to adopt the FM loss function as our dissimilarity metric $L$. As demonstrated by [6], the FM loss function provides an upper bound on the $W_2$ distance between the target density and the pushforward density. Therefore, minimizing the FM loss function guarantees the feasibility criteria. The simulation-free training scheme of FM offers a computationally efficient method for pretraining boundary conditions and guaranteeing the feasibility of the boundaries throughout the density path optimization algorithm. In Figure 5a we show the pushforward of the spline obtained from initializing all the control points at **0**. In Figure 5b we show the resulting optimized path using the PDPO strategy. The running time for this case is 24 m 32s, leading to an action value of 40.31. Since the collocation points are initialized to **0**, we did not use a geodesic-warmup. Compare this solution with the one reported in Section 4.2.

### F.1.2  Reusability of boundary parameters

A crucial feature of PDPO is its boundary parameter reusability: the boundary condition models $\theta_0$ and $\theta_1$ can be trained independently of an action functional's interior dynamics. This independence enables flexible reuse of the boundary parameters.

Specifically, when solving multiple action minimization problems that share boundary constraints, PDPO offers the following flexibility: after training a library of generative models **with shared architecture** $\{\theta_{\rho_1}, \theta_{\rho_2}, \ldots, \theta_{\rho_k}\}$ for distributions $\{\rho_1, \rho_2, \ldots, \rho_k\}$, any action minimization problem

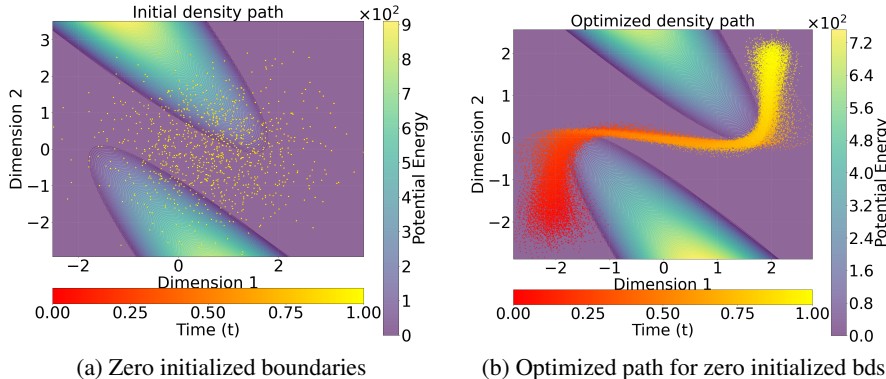

(a) Zero initialized boundaries      (b) Optimized path for zero initialized bds

Figure 5: No pertaining for $\theta_0$ and $\theta_1$.

Table 3: Comparison for the SCC problem. The quantities $W_2(\rho_i)$, $i = 0, 1$, denote the empirical Wasserstein-2 distances at times $t = 0$ and $t = 1$, computed using the POT library [12].

| Problem (method) | $\mathcal{A}[\rho_t]$ | $\int_0^1 \mathbb{E}_{\rho_t}\big[V(X)\big]\,dt$ | $(W_2(\rho_0),\, W_2(\rho_1))$ | Time |
|---|---|---|---|---|
| SCC (PDPO) | **432.23** | 3.42 | **(0.12, 0.11)** | **36 m 12 s** |
| SCC (GSBM) | $440.02 \pm 0.75$ | **3.12** | (0.59, 1.76) | 53 m 13 s |

with boundary conditions in this set can reuse the parameters of the pretrained models, regardless of the direction of transport or the specific action functional. For example, if Problem 1 transports from $\rho_i$ to $\rho_j$ and Problem 2 transports from $\rho_j$ to $\rho_i$, both problems can use the same pretrained pair $(\theta_{\rho_i}, \theta_{\rho_j})$ by simply swapping their roles as initial and terminal conditions.

This modularity significantly reduces computational cost when solving families of related problems. For instance, in the opinion depolarization example, once $\theta_0$ and $\theta_1$ are trained (29m35s and 31m12s, respectively), they can be reused for alternative action functionals, different polarization dynamics, or even reversed boundary conditions, provided the marginal distributions remain unchanged.

We treat the pretraining phase for the boundary models as a *one-time investment*.

To demonstrate this reusability feature, we reuse the pretrained boundary-condition models from the VNEFI example in a different setting, under a modified action functional and with reversed flow direction. In the VNEFI problem, the boundary conditions are

$$\rho_0 = \mu = \mathcal{N}\left(\begin{bmatrix} -11 \\ -1 \end{bmatrix}, 0.5I\right), \qquad \rho_1 = \nu = \mathcal{N}\left(\begin{bmatrix} 11 \\ 1 \end{bmatrix}, 0.5I\right).$$

Denote the corresponding pretrained models by $\theta_\mu$ and $\theta_\nu$ (training times: 0.38 s and 0.39 s, respectively).

We then apply these same boundary models to solve the **SCC problem** from [20], reversing the flow direction so that $\rho_0 = \nu$ and $\rho_1 = \mu$. Figure 6a shows the pushforward of the control points, Figure 6b the interpolated trajectory, and Figure 6c the corresponding GSBM solution. Quantitative results are provided in Table 3.

As before, PDPO achieves a lower action value and shorter training time than GSBM. It is important to emphasize that the boundary parameters are *reused* from the VNEFI example, so the cost of FM training is *amortized* across problems. Both methods yield similar constraint-violation levels, with GSBM showing a slightly smaller residual.

### F.2 Geodesic warmup

In the geodesic warmup, we optimize the linearly initialized control points using Algorithm 1 with $F(\rho) = 0$. This approach effectively reduces the computational cost of our algorithm. As we show in

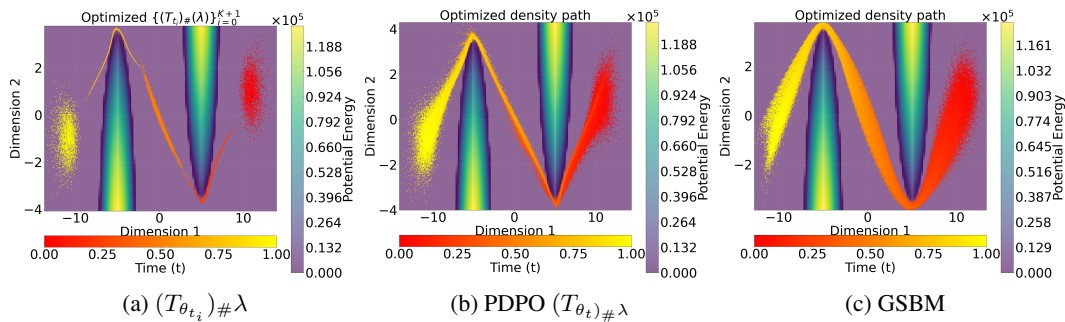

(a) $(T_{\theta_{t_i}})_{\#}\lambda$   (b) PDPO $(T_{\theta_t})_{\#}\lambda$   (c) GSBM

Figure 6: Comparison for SCC with new boundary conditions.

Figure 7, using a geodesic warmup drastically improves the speed of convergence and allows PDPO to reach a lower local minimum in the s-curve example.

To reduce the computational cost of the geodesic warmup, we consider at most $N = 15$ points for the trapezoidal rule. The solution computed without geodesic warmup took 5m 32ss, whereas the solution with geodesic warmup took 5m 57s. Thus, the computational overhead of the geodesic warmup is negligible.

In Figure 8, we show a comparison of a solution without geodesic warmup (Figures 8a and 8b) and with geodesic warmup (Figures 8 and 8d).

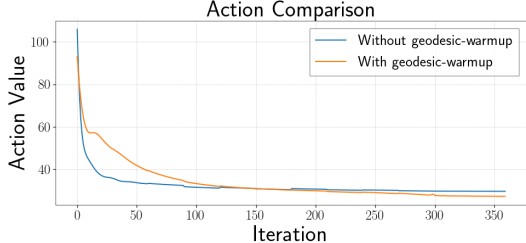

Figure 7: Comparison of action for solutions with and without geodesic warmup.

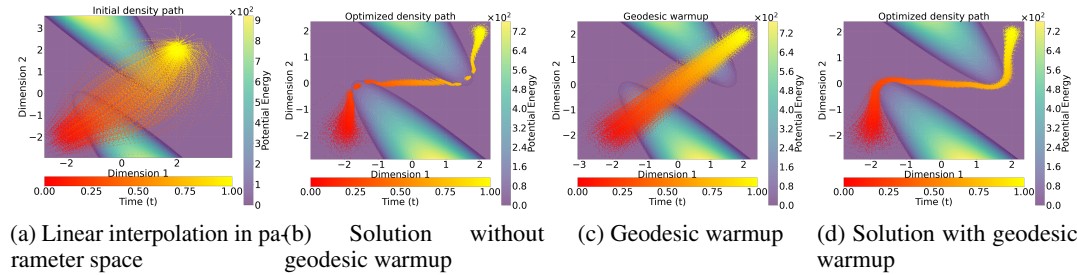

(a) Linear interpolation in parameter space   (b) Solution without geodesic warmup   (c) Geodesic warmup   (d) Solution with geodesic warmup

Figure 8: Comparison of solutions with and without geodesic warmup.

### F.3   Ablation Control Points

In Figure 9 we can see the comparison of the difference in the solution obtained by PDPO when varying the number of control points $K$. As we can see, when the number of control points increases, the quality of the solution also increases. Observe that the transition between the obstacles is smoother as the number of control points increases.

Table 4: Experimental set-up

| | GMM | V-neck | S-tunnel | Opinion | |
|---|---|---|---|---|---|
| d | 2 | 2 | 2 | 2 | 1000 |
| K | 5 | 3 | 5 | 3 | 3 |
| N | 30 | 60 | 30 | 20 | 20 |
| M | 1000 | 1000 | 1000 | 1000 | 5000 |
| Architecture | [2,256,4] | [2,128,4] | [2,64,4] | [2,128,4] | [1000,128,4] |
| Epochs | 15 | 15 | 18 | 10 | 10 |
| Coupling opt steps | 20 | 20 | 20 | 20 | 20 |
| Path opt. steps | 30 | 20 | 30 | 20 | 20 |
| Geodesic warmup steps | 100 | 100 | 100 | 200 | 200 |
| $\alpha$ | 100000 | 100000 | 100000 | 100000 | 10000 |
| $(\kappa_0, \kappa_1, \kappa_2)$ | (50,0,0) | (3000,50,0) | (100 , 0 ,5) | (0,50,0) | (0,50,0) |
| Mean $\rho_0$ | $e^{16i\pi}, i=0,\dots,7$ | $\begin{bmatrix}-11\\-1\end{bmatrix}$ | $\begin{bmatrix}-2\\-2\end{bmatrix}$ | **0** | **0** |
| Mean $\rho_1$ | $e^{8i\pi}, i=0,\dots,3$ | $\begin{bmatrix}11\\1\end{bmatrix}$ | $\begin{bmatrix}2\\2\end{bmatrix}$ | **0** | **0** |
| Covariance of $\rho_0$ | **I** | 0.5**I** | 0.1**I** | $\text{diag}(\begin{bmatrix}0.5\\0.25\end{bmatrix})$ | $\text{diag}(\begin{bmatrix}4\\0.25\\\vdots\\0.25\end{bmatrix})$ |
| Covariance of $\rho_1$ | **I** | 0.5**I** | 0.01**I** | 3**I** | 3**I** |

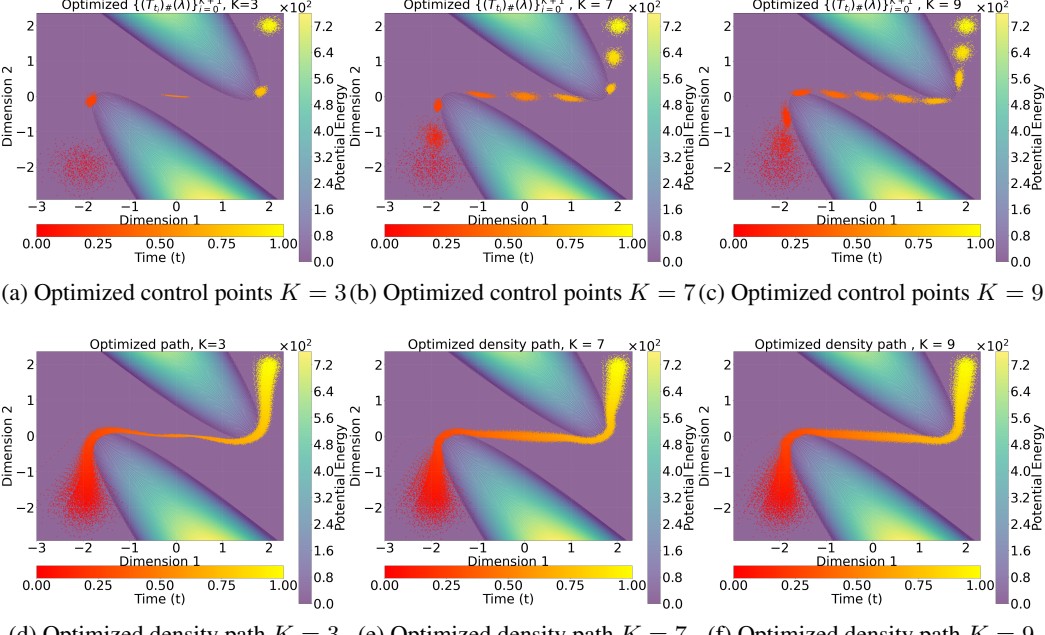

(a) Optimized control points $K = 3$   (b) Optimized control points $K = 7$   (c) Optimized control points $K = 9$

(d) Optimized density path $K = 3$   (e) Optimized density path $K = 7$   (f) Optimized density path $K = 9$

Figure 9: Comparison of solutions with increasing number of control points $K$.

## F.4   Obstacle Avoidance with Mean Field Interactions

Here we provide the definitions for the entropy $\mathcal{E}(\rho)$ and congestion potential $\mathcal{C}(\rho)$,

$$\mathcal{E}(\rho) = \int_{\mathbb{R}^d} \log(\rho)\rho dx \quad \text{and} \quad \mathcal{C}(\rho) = \int_{\mathbb{R}^{d \times d}} \frac{2}{\|x - y\|^2} \rho(x)\rho(y) dx dy.$$

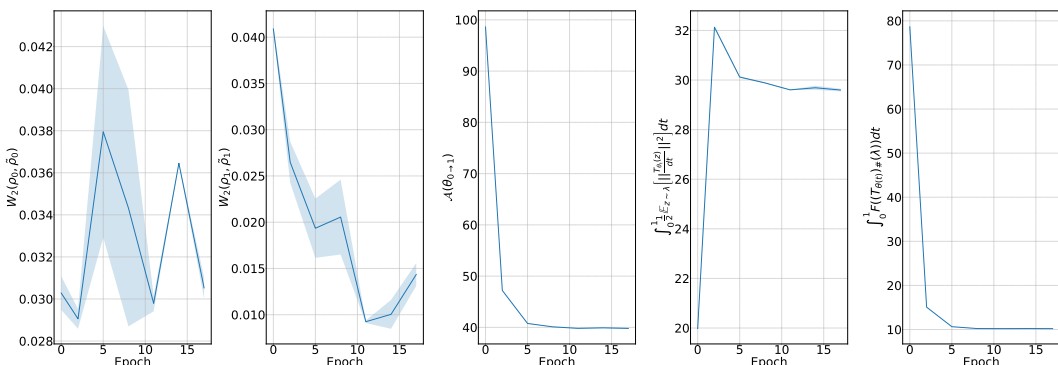

Figure 10: Quantities of interest with uncertainty estimates along the training process, **S-curve**.

**Remark:** In Table 1 we reported the $W_2$ distance of the boundaries for GSBM. The samples at the terminal time $t = 1$ are generated by the forward solver, while the samples at the initial time $t = 0$ are produced by the backward solver.

**S-curve** The definition of the problem and source code were taken from [17]. We refer to it for the definition of the obstacle. APAC-Net was more sensitive to the coefficient $\kappa_0$ and $\kappa_2$,; there we used $\kappa_0 = 5$ and $\kappa_2 = 1$.. The action reported in Table 1 for APAC-Net was obtained by evaluating its solution using our code with the values of $\kappa_0, \kappa_2$ reported in 4. In Figure 10 we show the $W_2$ distance with the boundaries, action, kinetic energy, and potential function along the training process. In these plots, an epoch is the completion of coupling + path optimization iterations. In these plots, we can clearly see that the feasibility condition is met from epoch 0 thanks to our pre-training strategy.

Both schedulers in this experiment are StepLR. The learning rate for the coupling optimizer is $10^{-4}$ with a step size of 10, and $\gamma = 0.9$. The learning rate for the path optimizer is $5 \times 10^{-4}$, step size of 10 and $\gamma = 0.1$.

**V-Neck** The definition and source code were taken from [20]. When GSBM's training time is constrained to match PDPO's training time, its action value is the same as before, but the boundary approximation is noticeably worse, $W_2(\rho_0, \tilde{\rho}_0) = 0.12 \pm 0.002$, $W_2(\rho_1, \tilde{\rho}_1) = 0.107 \pm 0.002$ 1h 15m 24s.

In Figure 12 we report the quantities of interest along the training epochs. See 11 to see the pushforward of the three control points, and the comparison of the PDPO and GSBM solutions.

Both schedulers in this experiment are StepLR. The learning rate for the coupling optimizer is $10^{-4}$ with a step size of 5, and $\gamma = 0.1$ The learning rate for the path optimizer is $5 \times 10^{-3}$, step size of 5 and $\gamma = 0.25$

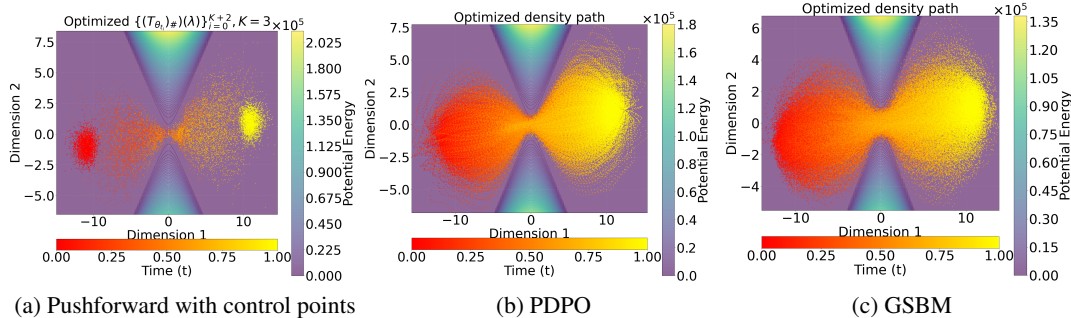

(a) Pushforward with control points     (b) PDPO     (c) GSBM

Figure 11: V-neck-E-FI a) Pushforward with control points. b) Pushforward with interpolated curve. c) Solution by GSBM.

**GMM** The definition and source code were taken from [20]. See Figure 13 for the comparison of the solutions. In this example, we use $\lambda = \rho_0$. To guarantee this is satisfied, we define $\theta_0 := \mathbf{0}$, the zero

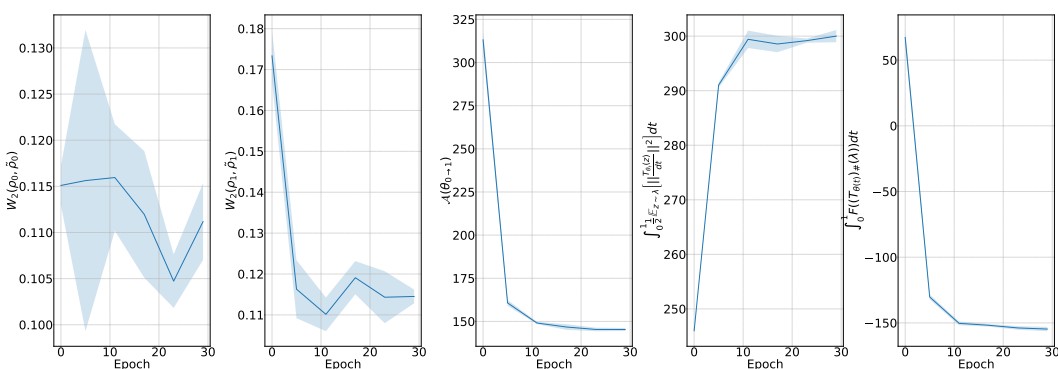

Figure 12: Quantities of interest with uncertainty estimates along the training process, **vneck**

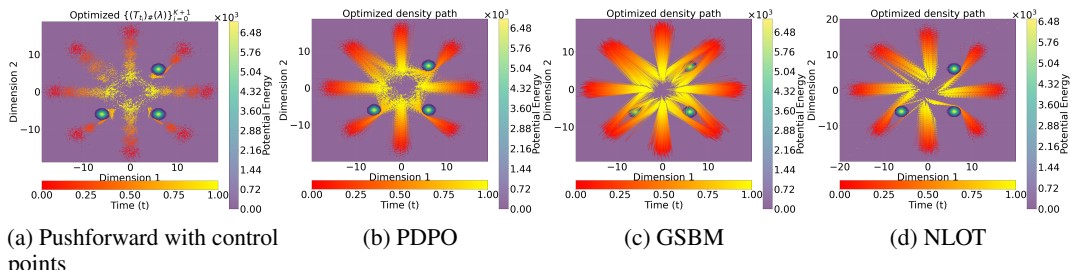

(a) Pushforward with control points

(b) PDPO

(c) GSBM

(d) NLOT

Figure 13: Comparison for GMM example

vector. In Figure 14 we report the quantities of interest along the training epochs. See 13 to see the pushforward of the five control points, and the comparison of the PDPO and GSBM solutions.

The schedulers in this experiment are cosine for the coupling optimization and StepLR for the path optimization. The learning rate for the coupling optimizer is $5 \times 10^{-6}$, the setup for the cosine scheduler is $T_0 = 5$, $T_{\text{mult}} = 2$, $\eta_{\min} = 1 \times 10^{-6}$. The learning rate for the path optimization is 0.001, step size of 3, and $\gamma = 0.9$

### F.5 Opinion Depolarization

In this problem, opinions $X(t) \in \mathbb{R}^{1000}$ evolve according to a polarizing dynamic:

$$\frac{dx(t)}{dt} = f_{\text{polarize}}(x(t); p_t),$$

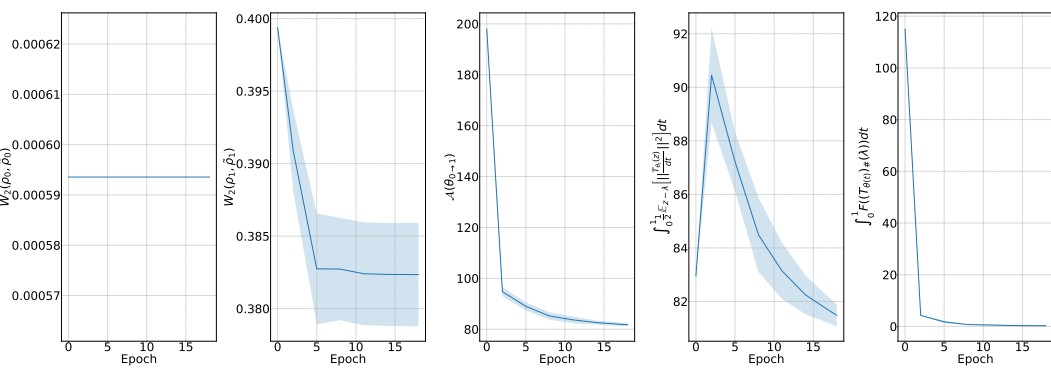

Figure 14: Quantities of interest with uncertainty estimates along the training process, **GMM**

with

$$f_{\text{polarize}}(x; p_t, \xi_t) := \mathbb{E}_{y \sim p_t}[a(x, y, \xi_t)\bar{y}], \quad a(x, y, \xi_t) := \begin{cases} 1 & \text{if } \text{sign}(\langle x, \xi_t \rangle) = \text{sign}(\langle y, \xi_t \rangle) \\ -1 & \text{otherwise} \end{cases}.$$

When this dynamic evolves without intervention, opinions naturally segregate into groups with diametrically opposed views. However, the desired outcome is a unimodal distribution.

To solve this problem, we follow [20] and incorporate the polarizing dynamics as a base drift or prior. See [11] for a reference in OT problems with prior velocity fields. Specifically, the action to optimize in the parameter space is defined by

$$\mathcal{A}_{\text{polarize}}(\theta_{0 \to 1}) := \mathbb{E}_{z \sim \lambda}\left[\frac{1}{2}\|f_{\text{polarize}}(T_{\theta(t)}(z); (T_{\theta(t)})_{\#}(\lambda)) - \frac{d}{dt}T_{\theta(t)}(z)\|^2\right] + \int_0^1 F((T_{\theta(t)})_{\#}(\lambda))dt.$$

The potential energy term is a congestion cost that encourages particles to maintain a distance from each other. We follow the experimental setup in [20] and [19]. Because of the dimension, we can only simulate deterministic dynamics, as specified in [30]. In Figure 15 we compare the optimized density trajectory and directional similarity histogram. These plots show that both methods obtain a nonpolarized trajectory.

### F.6  50D Scrhödinger Bridge

The SB between Gaussians has a closed-form solution [8]. In Table 5, we evaluate PDPO's accuracy by computing two different path discrepancies: $\int_0^1 W_2^2(\rho_{\theta(t)}, \rho(t))dt$ and $\int_0^1 W_2^2(\rho_{\theta(t)}, \tilde{\rho}(t))dt$. Here, $\rho(t)$ represents the theoretical SB solution between the original Gaussian boundaries, $\tilde{\rho}(t)$ is the SB solution between the approximated boundaries $(T_{\theta_0})_{\#}\lambda$ and $(T_{\theta_1})_{\#}\lambda$, and $\rho_{\theta(t)}$ is the density path recovered by PDPO. The first integral measures how closely PDPO approximates the true SB solution, while the second evaluates how well PDPO solves the boundary-approximated problem.

While the tabular results quantify the global accuracy of our method, Figure 16 provides a more intuitive visualization through a 2D projection of sample trajectories. In the figure, blue points represent samples from our PDPO solution $\rho_{\theta(t)}$, green points show the theoretical solution $\rho(t)$, and red points display the boundary-matched solution $\tilde{\rho}(t)$. This visualization demonstrates that PDPO accurately approximates individual particle trajectories throughout the transport process, confirming the effectiveness of our approach even in high-dimensional settings.

Table 5: SB between Gaussians

| d | $\sigma$ | $\int_0^1 W_2^2(\rho_{\theta(t)}, \rho(t))dt$ | $\int_0^1 W_2^2(\rho_{\theta(t)}, \tilde{\rho}(t))dt$ |
|---|---|---|---|
| 50D | 1 | $0.778 \pm 0.0018$ | $0.789 \pm 0.001$ |

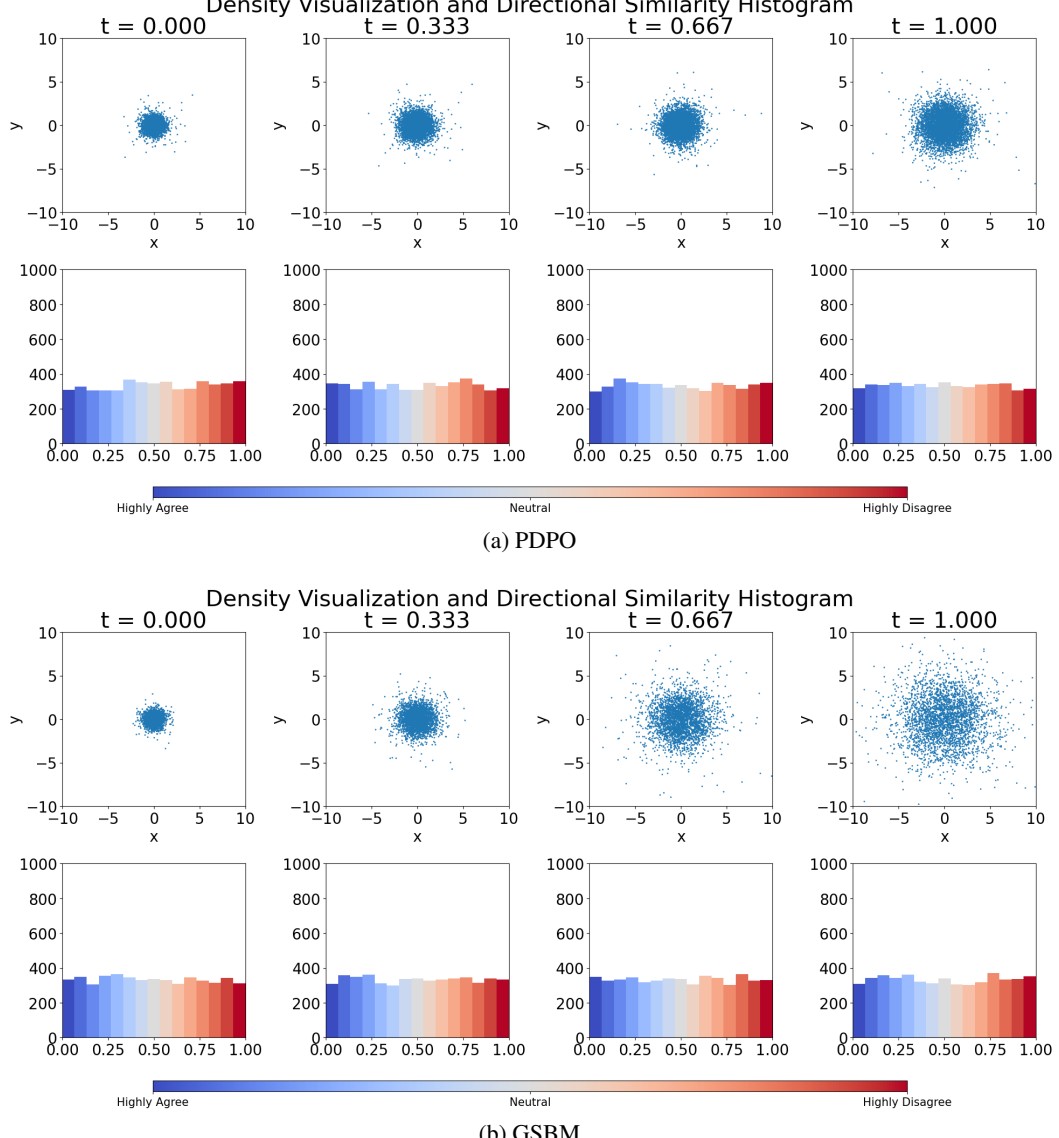

Figure 15: Comparison of optimized opinion polarization dynamics.

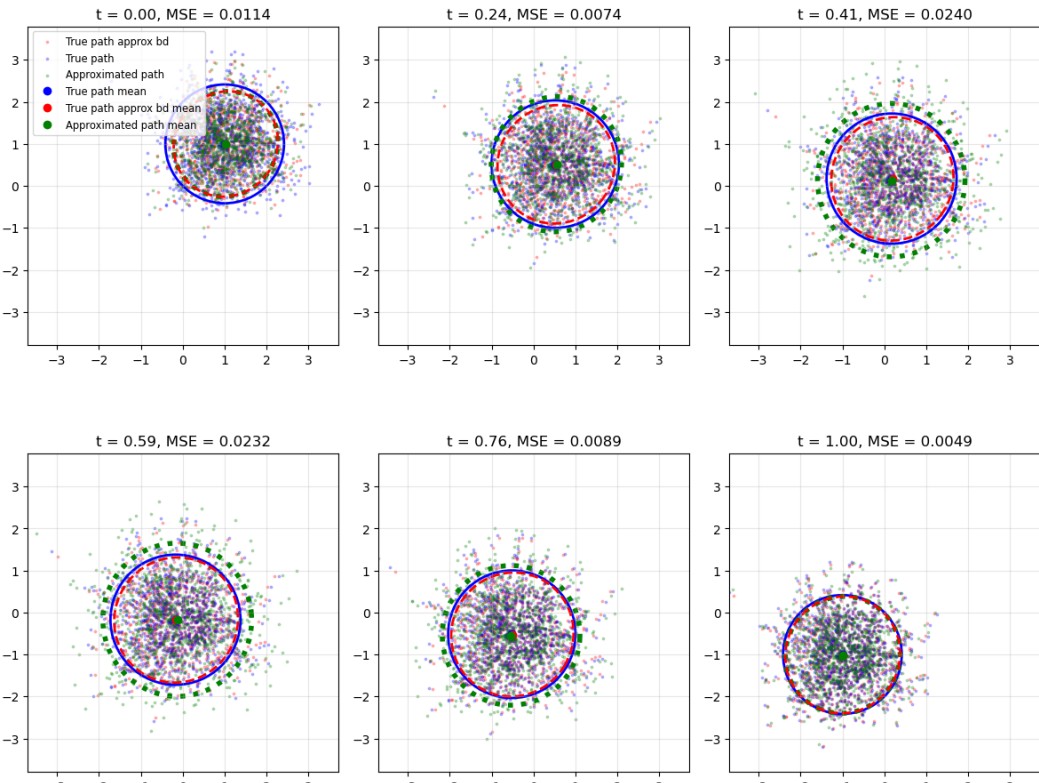

Figure 16: Particle comparison in two random directions.