# OpenReview forum: "PDPO: Parametric Density Path Optimization"
_NeurIPS.cc/2025/Conference — NeurIPS 2025 poster_

### Official Review · Reviewer_Q47s · 2025-07-01

**Clarity:** 3
**Significance:** 2
**Originality:** 2
**Rating:** 3
**Confidence:** 4

**Summary:**

This paper introduces Parametric Density Path Optimization (PDPO), a framework that casts the infinite‑dimensional least‑action problem over density curves into a finite‑dimensional spline optimization in the parameter space of a push‑forward map. The authors select neural ODEs for the map, discretize the parameter curve with cubic Hermite splines, and prove that using only 3 – 5 control points suffices to keep the action error $O(h^{\kappa-1})$.  Optimization proceeds by alternating a “path step’’ on interior control points with a “boundary step’’ on the endpoints (Algorithms 1‑3).  Experiments on obstacle‑avoidance transport, generalized momentum minimization, acceleration‑regularized dynamics, and a 1000‑d opinion‑depolarisation benchmark show 7–39 % lower action and up to 10× faster wall‑time than GSBM, NLOT and APAC‑Net in the authors’ setup.

**Questions:**

- Is it possible to  provide theoretical guarantees for the convergence of the alternating optimization procedure? Under what conditions does the bilevel optimization converge to a global or local optimum? The current analysis is insufficient for a rigorous optimization method.
- How sensitive is the method to the initialization of boundary parameters? Can you provide a systematic analysis of when the Flow Matching initialization will succeed, and what happens when it fails to find good boundary parameters?
- Computational Complexity: The $O(N \times M \times (1 + d))$ scaling for Fisher information terms seems prohibitive for truly high-dimensional problems. Can you provide a more detailed computational analysis and comparison with baselines using the same computational resources and implementation quality?
- Can you provide more rigorous statistical analysis including confidence intervals, significance tests, and fair computational comparisons? The current evaluation with only three seeds and potentially unfair runtime comparisons undermines the claimed performance improvements. Please also answer the point raised on taking initialization into runtime as I raised above.
- What are the precise assumptions required for the two theorems , and how realistic are they in practice? When do the error bounds become meaningful, and how do they relate to the quality of the final solution?

**Ethical Concerns:**

["NO or VERY MINOR ethics concerns only"]

**Final Justification:**

I think that the rebuttals (include ones to other reviewers) reveal substantial gaps in the original submission. The missing theoretical machinery, and implementation details about parametric MLPs represent fundamental content needed for understanding your work. Furthermore, while the 1000D opinion problem provides some evidence of scalability, the method remains limited to 2D toy problems and 50D Gaussians, with acknowledged architectural restrictions to manually-implemented MLPs. Notably, GSBM demonstrates image transport capabilities (AFHQ dataset, Section 4.2) that PDPO cannot handle due to your architectural limitations. This selective comparison on problems favorable to the method's constraints undermines claims of general applicability.

Given these clarifications, I raise my score from 2 to 3 (Borderline Reject). I think the core ideas have merit but require substantial revision before publication.

**Limitations:**

Yes but not enough.

**Paper Formatting Concerns:**

No.

**Quality:**

2

**Strengths And Weaknesses:**

- The paper addresses an important problem in optimal transport and provides a computationally efficient alternative to existing methods that struggle with high-dimensional settings.
- I think the paper’s strongest point is the clean static‑coupling reformulation (Eq. 11) and its parametric version (Eq. 13) that lets one exploit off‑the‑shelf autodiff once everything lives in the parameter space. The spline error bound is convenient and, to my knowledge, new for neural push‑forward families.
- Empirically PDPO is much faster than GSBM on the same GPU while matching or improving action and endpoint fidelity, especially in the 1 000‑d test where runtime drops from five hours to 23 minutes.

Quality‑wise, however, several issues remain:

- First, the dynamic/static equivalence (Theorem 1) merely states without the assumptions on the regularity of $\pi$. Moreover, the non‑emptiness of $\Gamma(\pi)$ looks restrictive but is not discussed in the main text, so I am not sure how often the theory covers the practical neural‑ODE setting. There are also no convergence guarantees for the proposed bilevel optimization procedure, which alternates between updating interior control points and boundary parameters, which I think is critical.

- Second, the experimental evaluation, while showing improvements, has notable limitations. Many experiments only compare against GSBM, and the baseline comparisons may not be entirely fair due to different implementations and computational frameworks. The statistical analysis is minimal - using only three random seeds.

- Third, from a technical standpoint, the method requires careful initialization of boundary parameters, and the authors acknowledge this is done "outside of the main algorithm." This suggests the approach may be sensitive to initialization and require problem-specific tuning. The scalability analysis reveals concerning computational complexity of $O(N \times M \times (1 + d))$ for problems involving Fisher information terms, which severely limits applicability to high-dimensional problems.

- Fourth, I find this statement in Section 3.3 troublesome: **"We in general consider this step outside of the main algorithm. A parameter can be used for any functional, so once a boundary parameter has been obtained, it can be reused across multiple problems."** I think this makes the runtime comparison completely unfair.
- Finally, the paper also lacks important details about convergence criteria and stopping conditions for the alternating optimization procedure. The claim that "as long as the action is lower bounded, step 2 will converge to a local minimum" is insufficiently rigorous without proper analysis of the optimization landscape.

- Clarity-wise, the authors have done a decent job. Just one point that I am not a big fan of putting the algorithm boxes in the Appendix, especially with the split of supplementary material for NeuRIPS submission.

---

> ### Author Rebuttal · Authors · 2025-07-31
>
> We appreciate the detailed review, but several fundamental misunderstandings about our method and factual errors about our experimental setup require correction.
>
> **Error bound claim**: The reviewer states we "prove that using only 3–5 control points suffices to keep the action error $O(h^{k-1})$." This is **not** what we claim. Our theoretical analysis (Theorem 4, Appendix C) shows that **if** the path $\theta(t)$ is $C^l$ with $l \geq 2$, then the action error scales as $O(h^{\kappa-1})$, where $\kappa = \min(l,4)$. Separately, we empirically observe that **3–5 control points suffice in practice** to capture high-quality paths across diverse problems. These are distinct claims - one theoretical, one empirical.
>
> **Dynamic/static equivalence assumptions**: Theorem 1 **does not** require regularity assumptions on $\pi$. **As shown** in lines 137-139, we guarantee $\tilde{\Pi} \neq \emptyset$, it contains optimal transportation coupling $\pi^{\*}$ as the McCann interpolation is a diffeomorphic flow. Since **the traditional Kantorovich problem has no regularity requirements** on $\pi$, neither does our static formulation.
>
> **Non-emptiness of $\Gamma(\pi)$**:  **We addressed this** for the Kantorovich coupling $\pi^{\*}$ in lines 137-139. We add a detailed discussion for a general $\pi = (id,T_\pi)\\#(\rho_0).$ If we assume $T_\pi$ is a diffeomorphism, then $\Gamma(\pi) \neq \emptyset$ iff the identity map $id$ and $T_\pi$ are smoothly isotopic[1]. Two diffeomorphisms $f_0, f_1$ are smoothly isotopic if there exists a smooth homotopy $\\{f_t\\}_{0 \leq t \leq 1}$ where each $f_t$ is a diffeomorphism [1]. By Lemma 2, Chapter 6 [2], a map $T$ is smoothly isotopic to $id$ if $\det(DT) > 0$ everywhere.
>
> We define:
> $$\mathcal{T}(\rho_0,\rho_1) = \\{T:\mathbb{R}^d \to \mathbb{R}^d : T \text{ diffeomorphic}, T\\#(\rho_0) = \rho_1, \det(DT) > 0\\}$$
>
> Since the Monge map $T^{\*}$ satisfies $\det(DT^{\*}) > 0$, we have $\mathcal{T}(\rho_0,\rho_1) \neq \emptyset$. We modify $\tilde{\Pi}$ to:
> $$\tilde{\Pi} = \\{\pi \in \Pi(\rho_0,\rho_1): \pi = (id,T_\pi)\\#(\rho_0), T_\pi \in \mathcal{T}(\rho_0,\rho_1)\\}$$
>
> The proof of Theorem 1 remains unchanged with this new definition.
>
> We thank the reviewer for raising this point. Providing a rigorous discussion of the non-emptiness of $\Gamma(\pi)$ allowed us to make our theoretical framework more precise, and we appreciate the opportunity to clarify this subtle but important assumption.
>
> ***Neural ODE coverage**: The reviewer appears to misunderstand how NODEs are used in our framework. **Clarification** NODEs are used to define the parametric maps $T_\theta$. Flows/paths are approximated by the curves $T_{\theta(t)}(z)$ for $z \sim \lambda$, $t \in [0,1]$.
>
> **Convergence guarantees**:  The reviewer demands convergence analysis for our bilevel optimization. We have two separate problems:
> 1. Density space (11): For linear coercive actions, [4] Chapter 7 shows unique transport plans (Corollary 7.23)
> 2. Parameter space (13): For linear coercive actions, we expect global minimizers given sufficient parametric expressiveness.
> Convergence analysis for non-linear potentials represents a fundamental open challenge affecting all neural approaches in optimal transport, not a limitation specific to our method.
> Numerically, our convergence tables show consistent action reduction:
> | Bilevel optimization cycles | Action value $\mathcal{A}$ |
> | -------------------- | ------------- |
> | 0   | 18562.34 |
> | 7   |     476.81 |
> | 15   |  473.19  |
> | 23    |    471.40  |
> | 31    |   471.71    |
>  We currently use maximum iterations as the stopping criteria. Common alternatives in bilevel optimization include gradient norm thresholds or loss change thresholds. Testing these could improve convergence detection and efficiency. In Figures 9-13, **we demonstrated** consistent action convergence across experiments with uncertainty bounds.
>
> **Baseline comparisons**: We compare NLOT and APAC-Net **when** applicable.  NLOT is **limited** to linear potentials while APAC-Net uses soft terminal costs instead of hard distributional constraints. GSBM is the **only** baseline capable of handling all settings we consider.  GSBM is the **SOTA**, the only method with equivalent flexibility to ours. The only other comparable method is DeepSBM [4], which GSBM already outperforms.
>
> **Implementation and computational fairness**: As stated in the paper we use **official implementations** with **recommended parameters** on identical hardware (AMD 7543 CPU + NVIDIA RTX A6000 GPU). To explore the computational fairness question, we compare the architectural differences:
> | Method | GMM | S-Curve | V-Neck | Opinion 2D | Opinion 1000D |
> |--------|-----|---------|--------|------------|---------------|
> | GSBM | 83,202 | 83,202 | 83,202 | 1,152,770 | 1,664,744 |
> | PDPO | 133,122 | 8,706 | 33,794 | 33,794 | 290,280 |
>
> PDPO achieves superior performance with simpler and often smaller networks.
>
> To test whether parameter count explains PDPO's speed advantage, we solve S-Curve ($\sigma = 0$) (with the GSBM setup) and V-Neck ($\sigma = 1$) examples using **identical MLP  architectures** [2+1,128,4] for both methods. Both problems use $\rho_0 = \mathcal{N}(\begin{bmatrix} -11 \\ -11 \end{bmatrix},0.5I)$ and $\rho_1 = \mathcal{N}(\begin{bmatrix} 11 \\ 11 \end{bmatrix},0.5I).$ To show that our initialization scheme is agnostic to the functional, we use the same pretrained boundary parameters $\theta_0, \theta_1$ in both problems (training times: 0.33s and 0.35s, respectively).
>
> | Problem | Method | Time | $\mathcal{A}$ | $W_2^2(t=0)$ | $W_2^2(t=1)$ |
> |---------|--------|------|-------------|--------------|--------------|
> | S-Curve | PDPO | 12m34s | 471.86 | 0.099 | 0.115 |
> | S-Curve | GSBM | 25m35s | 485.69 | 1.816 | 1.835 |
> | V-Neck | PDPO | 1h10m | 148.91 | 0.085 | 0.081 |
> | V-Neck | GSBM | 1h16m | 153.80 | 0.192 | 0.283 |
>
> With identical architectures, PDPO still outperforms GSBM in both action value and boundary approximation quality, demonstrating that performance differences are not due to parameter count alone. This also illustrates boundary parameter reusability across different problems. Addressing the reviewer's fourth point.
>
> **Flow Matching initialization costs**: The complete FM training times are:
> | Distribution | Epochs | Time |
> |-------------|--------|------|
> | 2D Gaussians | 1k | ~40s each |
> | GMM | 2k | 3m10s |
> | 1000D Opinion | 4k | ~30m each |
> Including FM training time, PDPO still outperforms baselines significantly.
>
> **Boundary parameter sensitivity**: Once trained, boundary parameters **are robust and reusable across problems with different potentials**, as shown in this rebuttal. We treat initialization as "outside" because it's **general** and **amortized**. The same $\theta_0$ can be reused for multiple optimization problems, as demonstrated in the table showing the performance of the models with the same architecture. We emphasize this is a feature, not a limitation. Furthermore, Appendix F.1 demonstrates successful optimization starting from $\vec{0}$ parameters, achieving action value 40.31 in 24m32s without geodesic warmup. This shows our method's robustness to initialization.
>
> ## Computational Complexity & Limitations
>
> **Fisher Information scaling**: The $O(N \times M \times (1+d))$ complexity for Fisher Information is fundamental to computing Fisher Information in any method, not unique to PDPO. As future work, we plan to explore Hutchinson-type estimators, see [5] for $\nabla \log(\rho)$ to address this limitation.
>
> **Statistical significance**: Three seeds are a standard practice in this field (GSBM, DeepSBM). Additionally, our convergence curves with uncertainty estimates across optimization iterations provide strong evidence.
>
>
> **Flow Matching success conditions**: Flow Matching is SOTA in generative modeling [6], so we expect to find good boundary parameters with appropriate tuning. Our boundary optimization combines FM loss with action minimization, maintaining generation quality while improving coupling optimality. Furthermore, the FM component of our algorithm can be replaced with other models such as Stochastic Interpolations [7].
>
> **Theorem assumptions**: **All assumptions are explicitly** stated in Appendix C, Theorem 4. We detail smoothness requirements, Lipschitz continuity, bounded derivatives, and their relationship to action error bounds. We also provide corollaries for spline interpolation and Wasserstein error bounds.
>
> [1] Introduction to Differential Topology Joel W. Robbin and Dietman A. Salamon. Lecture course “Differential Geometry II” held by the
> second author at ETH Z¨urich in the spring semester of 2018.
> [2] From the Differentiable Viewpoint John W. Milnor. The University Press of Virginia, Charlottesville (1981).
> [3] Villani
> [4] Deep Generalized Schrödinger Bridge. Guan et al. 36th Conference on Neural Information Processing Systems
> [5]FFJORD: FREE-FORM CONTINUOUS DYNAMICS FOR SCALABLE REVERSIBLE GENERATIVE MODELS. Will Grathwohl et al. ICLR 2019
> [6] Flow Matching Guide and Code. Yarlon Lipman et al. Nips 2024
> [7]Stochastic Interpolants: A Unifying Framework for Flows and Diffusions CoRR 2023

---

> ### Author Response · Authors · 2025-08-05
>
> Thank you for acknowledging that our rebuttal was comprehensive and addressed several of your concerns. However, we respectfully disagree with the characterization that our responses "reveal substantial gaps in the original submission."
> Regarding "Missing Content":
>
>    - **Theoretical machinery**: If the reviewer refers to the use of isotopic diffeomorphism with the additional constraints $Det(T)>0$ and diffeomorphic $T$ in the definition $\tilde{\Pi},$ we included these **in response to your specific question** on the non-emptiness of $\Gamma(\pi)$ for **non-Kantorovich** $\pi$. The **original paper already** established this non-emptiness result (lines 137-139) for the Kantorovich $\pi$ using McCann interpolation.
>   - **Implementation details**: If the reviewer refers to the parametric MLP explanation, these details **were included in Appendix E, and the associated code was submitted.**
>
> - **Problem Selection and Capabilities**:
>      -  The reviewer **underplays the significance** of the 1000D opinion dynamics benchmark. To the best of our knowledge, only GSBM and DeepGSB can handle this problem in 1000D. The problem represents a substantial computational challenge.
>      -   The reviewer focuses on our inability to handle images while **not fully acknowledging** that we've introduced a **new problem class**: Acceleration-regularized dynamics with potential terms. We demonstrate the first method capable of solving momentum problems with potential terms, extensible to arbitrary p-th derivative regularization, a capability that existing methods don't currently possess.
>     -  Image transport with potential-energy terms is a complex problem that required **several** iterations of modeling and refinement. GSBM **builds** on DeepGSB, **sharing** core authorship, code structure, and the forward/backward SDE framework for density control. DeepGSB **does not** include any image-domain experiments, highlighting that image transport capabilities were introduced only through subsequent development in GSBM. **PDPO represents the first parametric approach to action-minimizing problems with hard boundary constraints**. That a novel algorithmic/theoretical framework can achieve SOTA results on most benchmarks, while simultaneously opening new problem domains, should be included in this developmental context.
>
> - **Regarding Architectural Limitations:** Current MLP constraints don't invalidate our fundamental theoretical contributions: static reformulation, parameter space optimization, and bilevel framework. These advances are architecture-agnostic. In future work, we plan to exploit JAX/Flax functional programming design, which will remove the MLP constraints imposed by PyTorch's object-oriented design.
>
> In summary, we've introduced a new framework, we've explored new problem formulations, while achieving computational efficiency gains for benchmark problems. We consider PDPO to be a **strong first step** for **parametric-based approaches** in the density control problems with fixed boundary conditions domain.
>
> We appreciate the score improvement and feedback that will strengthen the final manuscript. We will **certainly** incorporate the valuable feedback from all reviewers and rebuttal discussions into the final manuscript

---

### Official Review · Reviewer_Nr1t · 2025-07-02

**Clarity:** 3
**Significance:** 2
**Originality:** 3
**Rating:** 4
**Confidence:** 2

**Summary:**

The proposed method, called Parametric Density Path Optimization (PDPO), utilizes an action minimization framework incorporating additional constraints such as external potential energy and interaction and internal energy to find an action-minimizing path between probability densities. This generalizes the classical approach used in the case of Wasserstein distances which can also be seen via the action-minimizing viewpoint.

The authors work with a parameterization which allows for a finite dimensional representation making the action minimization problem computationally tractable. The authors furthermore present a theorem establishing the equivalence between a  static and dynamic representation viewpoint of their constrained action-minimization approach (Theorem 1).

A cubic Hermite spline approximation of parameter space curves together with a neural ODE approach is then used to solve the optimization problem practically and an error estimate for the spline-based approximation is provided (Theorem 2).

Experiments are presented that include two-dimensional settings of obstacle avoidance problems, showing improved action cost and significantly reduced computation time vs baseline methods.
The authors also extend their approach to incorporate an acceleration term and show 2d examples where PDPO can find solutions also in this setting. Additionally, a 1000-dimensional case of opinion depolarization problem is considered where the method again has improved computational time as compared to the baseline GSBM approach.

**Questions:**

- The current 2D maps you considered still have a relatively simple spatial structure. Do you think your method will generalize to complex maps directly, or do you see challenges in convergence as problem complexity is increased?

**Ethical Concerns:**

["NO or VERY MINOR ethics concerns only"]

**Final Justification:**

The work makes a valuable contribution in my view. Following the rebuttal comments by the authors, I maintain a "borderline accept" recommendation primarily due to the lack of an additional more realistic/higher complexity experimental evaluation scenario.

**Limitations:**

- It would be interesting to add some discussions of potential failure cases of the proposed method as part of the limitations. An experiment where the approach could not be successfully applied yet or where other challenges of interest arise may be helpful to guide readers towards open problems within this area.

**Paper Formatting Concerns:**

No paper formatting concerns.

**Quality:**

3

**Strengths And Weaknesses:**

Strengths:
- overall, the paper is well-written and clearly describes the chosen problem and solution approach

- significant improvements in computational complexity and improvements or matching performance in minimization of the objective function over the baseline approaches appear empirically supported by the presented experiments.

- the work tackles a rather general action minimization path finding problem between probability densities of relevance to several application domains in the NeurIPS community.

- the paper presents both the authors' theoretical insights and formulation (e.g. Theorem 1 and 2) as well as a practical algorithm and implementation that combines a neural ODE formulation in combination with the proposed parameterize to arrive at a practically feasible solution.

Weaknesses:
- The chosen obstacle avoidance scenarios are interesting, but may at present still be quite far from the required real-world complexity to be useful in end-applications such as in robotics, where much larger maps with significant detail would likely have to be considered to be of practical relevance.
- To assess the practical significance of the proposed approach it may be helpful to showcase an experiment with much larger problem complexity as well.

---

> ### Author Rebuttal · Authors · 2025-07-31
>
> We thank the reviewer for their positive assessment of PDPO's theoretical contributions, computational efficiency, and practical relevance. Below, we address their constructive questions:
>
> # Q1, challenges for complex maps.
> For more complex environments, we recommend tuning the number of control points K and the number of discretization steps for computing the action, N. We usually try to keep these parameters as low as possible, as they increase the computational cost of our method.
>
> #Open problems and limitations
>
> - Intermediate constraints or Multi-Marginal Constraints:  Extending PDPO to enforce constraints at intermediate timepoints (beyond the two-endpoint setting) introduces new coupling structures in the variational formulation. We are currently exploring how to modify Theorem 1 accordingly, which would also affect how intermediate control points are updated.
> - "Overdamped Brownian Motion in a Force Field" Section 8 in [Chen, 2024] studies an energy-minimizing formulation of overdamped Langevin dynamics in force fields.
> - "Density Control With Multiple Species", see section 4 [2]. Coupled transport with cross-species interactions.
> - Optimal transport with nonlinear diffusion equations. We want to make PDPO  compatible with nonlinear mobility equations such as those arising in nonlinear Fokker–Planck-type models or porous medium equations, e.g., in [3]
> - The Fisher Information term appears in action functionals corresponding to entropy-regularized or stochastic control problems. In our current implementation, the main bottleneck lies in evaluating the right-hand side of Equation (20), which involves the divergence of the score function. In contrast to $\log(\rho)$, efficient Hutchinson-like estimators for $\nabla(\nabla\cdot v)$  are less developed. Exploring scalable approximations to this term—especially using trace estimators or score-based models—would help push the dimensional limits of PDPO for stochastic optimal control.
>
>
> [1]Density Control of Interacting Agent Systems, Yongxin Chen, IEEE TRANSACTIONS ON AUTOMATIC CONTROL, VOL. 69, NO. 1, JANUARY 2024.
>
> [2] Nonlinear mobility continuity equations and generalized  displacement convexity. J. A. Carrillo, S. Lisini, G. Savare , D. Slepcev. Journal of Functional Analysis Volume 258, Issue 4, 15 February 2010, Pages 1273-1309.
>
> [3] FFJORD: FREE-FORM CONTINUOUS DYNAMICS FOR SCALABLE REVERSIBLE GENERATIVE MODELS. Will Grathwohl et al.  ICLR 2019

---

> > ### Comment · Reviewer_Nr1t · 2025-08-05
> > **Rebuttal**
> >
> > I would like to thank the authors for their rebuttal response and would like to encourage you to include a discussion of real-world complexity challenges and scaling to real-world applications in the final version of the paper. I will maintain my borderline accept recommendation.

---

### Official Review · Reviewer_5bmt · 2025-07-04

**Clarity:** 3
**Significance:** 3
**Originality:** 3
**Rating:** 4
**Confidence:** 2

**Summary:**

This paper introduces Parametric Density Path Optimization (PDPO), a novel method for calculating the optimal, action-minimizing paths between different probability densities. The core innovation is to reframe this complex, infinite-dimensional problem into a more manageable, finite-dimensional one. It achieves this by representing the probability path as the transformation (or "pushforward") of a simple, fixed reference density through a parametric map, which is modeled by a Neural ODE. The path is then optimized by interpolating a small number of control points (typically 3-5) in the parameter space using cubic Hermite splines, which defines the continuous evolution of the density.

PDPO demonstrates superior performance compared to existing state-of-the-art methods across several challenging benchmark tasks, including obstacle avoidance, entropy-constrained transport, and high-dimensional opinion dynamics. The method is not only more computationally efficient, achieving 40-80% faster runtimes, but it also produces higher-quality solutions, with 7-39% lower action and up to 10 times better boundary accuracy. A key strength of PDPO is its flexibility, as it can be applied to a wide range of problems involving external potentials, mean-field interactions, and higher-order dynamics, such as acceleration minimization, within a single, unified framework.

**Questions:**

### **1. On the Source of Computational Efficiency**

Regarding the computational efficiency of PDPO compared to GSBM, could you clarify the contribution of the **Flow Matching (FM) initialization**? The paper notes that PDPO achieves significantly faster runtimes. Is it possible that this speed-up is largely attributable to the FM initialization step? To help disentangle the gains from the core PDPO framework versus the initialization strategy, it would be insightful to see a comparison where the GSBM baseline is also initialized using a similar FM-based approach.

***

### **2. Clarification on Constraint Violation in Figure 2**

In the obstacle avoidance experiment shown in Figure 2, the paper claims that both GSBM (2c) and APAC-Net (2d) violate the obstacle constraints, while PDPO effectively avoids them. However, on visual inspection, the optimized PDPO path (2b) also appears to have some samples crossing into the obstacle regions. Could you please provide a **quantitative measure of the constraint violation** (e.g., the percentage of the mass inside the obstacle) for all three methods? This would offer a more objective comparison of their performance on this task.

***

### **3. Request for Concrete Implementation Details**

To improve the paper's reproducibility and provide a clearer understanding of the methodology, could you offer more concrete implementation details or a working example for two key aspects of the framework?

* **Parameterization of the Pushforward Map**: While the paper states that the map $T_{\theta}$ is a Neural ODE, a more detailed description of the network architecture ($v_{\theta}$) and how its parameters $\theta$ are structured would be very helpful.
* **Flow Matching Initialization**: A step-by-step walkthrough of how Flow Matching is employed to initialize the boundary parameters $\theta_0$ and $\theta_1$ would greatly clarify the practical application of this crucial first step.

**Ethical Concerns:**

["NO or VERY MINOR ethics concerns only"]

**Final Justification:**

I understand the effort the authors made, but I believe this does not affect my final rating, and I decided to leave the score unchanged.

**Limitations:**

### Scalability in High-Dimensional Stochastic Optimal Control (SOC)

A primary limitation is the method's computational cost when dealing with problems that involve Fisher Information, which is used to solve Stochastic Optimal Control (SOC) problems.

* The cost to compute the action functional increases substantially with the problem's dimension, `d`.
* Calculating the Fisher Information requires solving a separate `d`-dimensional Ordinary Differential Equation (ODE).
* The authors state that a 50-dimensional Schrödinger Bridge was the "highest-dimensional case with Fisher Information we have successfully computed," highlighting a practical cap on its scalability for this class of problems.

***

### Architectural Restrictions

The current implementation of PDPO has architectural constraints that limit its range of application.

* The framework uses Neural ODEs (NODEs) built with Multilayer Perceptron (MLP) architectures.
* This reliance on MLPs restricts the method's direct use for certain data types, particularly images, where specialized architectures like U-Nets are more effective.
* This is a notable limitation compared to methods like GSBM, which can be applied to image-based transport problems.

***

### Methodological and Evaluative Limitations

The methodology and evaluation presented in the paper have some implicit limitations.

* **Dependence on Initialization:** The three-step optimization process relies on a strong initial approximation of the boundary distributions using Flow Matching (FM). While effective, this also means the final solution's quality is dependent on this prerequisite step.
* **Potential for Unfair Comparison:** The performance gains reported over baselines like GSBM could be partially attributed to the specialized FM initialization. The paper does not provide an ablation study comparing performance when baselines are also given the benefit of a similar initialization, which makes it difficult to isolate the gains from the core PDPO algorithm alone.
* **Convergence to Local Minima:** The optimization algorithm, being gradient-based, is guaranteed to converge to a local minimum, not necessarily a global one. The final solution is therefore sensitive to the initial path, again underscoring the importance of the initialization procedure.
* **Qualitative Evaluation:** In some experiments, such as obstacle avoidance, the claims of superiority are based on visual inspection of plots. The paper does not provide quantitative metrics for constraint violation (e.g., the amount of probability mass inside an obstacle), relying on a qualitative assessment that can be subjective.

**Paper Formatting Concerns:**

Not from my side

**Quality:**

3

**Strengths And Weaknesses:**

### Strengths

* **Novel and Theoretically-Grounded Formulation:** The paper introduces Parametric Density Path Optimization (PDPO), a novel method that reframes a complex infinite-dimensional optimization problem into a more tractable finite-dimensional one. The core strength lies in its static formulation of the action-minimizing problem, which the authors claim is novel for handling internal and interaction energy terms, unlike previous work. This new formulation is supported by theoretical guarantees, including a theorem proving its equivalence to the classical dynamic formulation (Theorem 1) and an error bound for the cubic spline approximation (Theorem 2).

* **Superior Performance and Efficiency:** The paper provides strong empirical evidence that PDPO outperforms state-of-the-art methods in benchmark tasks.
    * **Accuracy:** It achieves 7-39% lower action costs and up to 10x better boundary accuracy compared to baselines like GSBM and NLOT.
    * **Speed:** It demonstrates significantly faster runtimes, ranging from 40-80% on some tasks to over 10x faster in high-dimensional cases (e.g., nearly 2 hours for GSBM vs. under 10 minutes for PDPO in the GMM experiment, and over 5 hours vs. 23 minutes for the opinion dynamics problem).
    * **Simplicity:** A key advantage is its ability to resolve complex, high-dimensional problems accurately using only 3-5 control points for the spline interpolation.

* **Flexibility and Broad Applicability:** PDPO is designed as a general framework that can flexibly accommodate a wide variety of problems within a unified optimization scheme. It is shown to be effective for tasks involving:
    * External potentials and obstacles.
    * Stochastic Optimal Control (SOC) through a Fisher Information term.
    * Mean-field interactions and congestion.
    * Higher-order dynamics, such as acceleration-regularized problems, which the authors claim PDPO is the first method capable of solving in this context.

### Weaknesses

* **Potentially Unfair Comparison Due to Initialization:** The proposed optimization strategy begins with an efficient initialization of the boundary parameters using Flow Matching (FM). While effective, this raises a question about the fairness of the performance comparison. The significant speed-up over GSBM might be partially due to this specific initialization strategy rather than solely the core PDPO framework. The paper does not provide an ablation study or results for a GSBM baseline enhanced with a similar FM initialization, which would be necessary to isolate the true source of the efficiency gains.

* **Reliance on Qualitative Visual Evidence for Key Claims:** In the obstacle avoidance experiment (Figure 2), the paper claims that competing methods violate the constraints while PDPO does not. However, visual inspection of the PDPO result (Figure 2b) suggests that some samples may still lie within the obstacle. The paper does not provide a quantitative measure of constraint violation (e.g., the percentage of probability mass inside the obstacle) to substantiate its claim of superiority, relying instead on visual interpretation which can be ambiguous.

* **Acknowledged Scalability and Architectural Limitations:** The authors are transparent about certain limitations of the current method.
    * **Scalability in SOC:** The computational cost for problems involving entropy and Fisher Information increases substantially with the problem's dimension. The authors note that the highest-dimensional SOC problem they successfully computed was 50-dimensional, implying that scalability in this domain remains a challenge.
    * **Model Architecture:** The current implementation uses Neural ODEs with MLP architectures. This restricts its direct application to other data modalities, such as images, where methods like GSBM can use more suitable architectures (e.g., U-Nets). The authors explicitly state this as a limitation for image-based transport problems.

---

> ### Author Rebuttal · Authors · 2025-07-31
>
> Thanks for the detailed review and recognition of PDPO's novelty and flexibility!
>
> ## **Q1: On the Source of Computational Efficiency**
>
> Great question! Let us clarify the contribution of Flow Matching (FM) initialization and why our speedups are genuine.
>
> **PDPO algorithm does not rely on FM**, although we strategically selected it. Let's review what exactly is needed to setup PDPO, why FM was selected, and how the initialization works.
>
> PDPO solves the nested minimization problem:
> $$\inf_{(\theta_0,\theta_1)\in \Theta_{0}^1}\inf_{\theta_{0\to1} \in \Theta_{0\to1}}\mathcal{A}(\theta_{0\to1})$$
>
> The outer optimization requires parameters in the set:
> $$\Theta_0^1 =  \\{(\theta_0,\theta_1)\in \mathbb{R}^D\times \mathbb{R}^D: (T_{\theta_i})\\#,\lambda = \rho_i, i = 0,1\\}$$
>
> This is essentially a **generative modeling question**: given reference density $\lambda$, target dataset $\rho_0$, and mapping $T_{\theta}$, how do we train $T_{\theta}$ to generate samples of $\rho_0$?
>
> We don't need FM specifically - we need **any generative model** that:
> 1. Uses invertible mappings (NODEs) for computing $\log(\rho)$ and $\nabla_x\log(\rho)$
> 2. Has an efficient loss function to evaluate $(T_{\theta_i})\\#\lambda \approx \rho_i$
>
> Examples include Flow Matching, Stochastic Interpolants, Diffusion, CNFs, etc. We chose FM because it has a **simple and fast loss function** to evaluate during boundary optimization steps.
>
> **Key insight:** Once trained, the parameter $\theta_0$ representing density $\rho_0$ can be **reused across different action minimization problems** with different functionals $F$ and different terminal conditions $\rho_1$ **without retraining**.
>
> **Intialization strategy:** The reviewr asks if its possible to a initialize GSBM with a similar FM-based approach. The answer is yes.
>
> GSBM uses two NODEs, a forward and bacwkard that define vector fields that transport $\rho_0\to\rho_1$ and $\rho_1\to\rho_0,$ respectively. To use a FM like initalization. We propose to initialize the forward and backward velocity fields as pre-trained NODEs. We use the GSBM method/algorithm to train them with a zero potential function and $\sigma = 0.1.$ This reduces to train using Stochastic Interpolants. The initialization strategy was tested on the S-curve problem but there were no noticeable effects. We pretrained the network for approximately 40s each and verified that they sampled accurately the target density. The the training time and action value remain almost the same as before.
>
> To clarify the computational cost of the FM initialization, we report the FM training details. In the table GMM1 refers to $\rho_1$ in the GMM example. In that case $\lambda = \rho_0$ so no training is needed
>
> | $\rho$       | T      |
> | -------------------------------------------------------------------------------------------------- | ------ |
> | $\mathcal{N}(\begin{bmatrix} -2 \\ -2 \end{bmatrix},0.1I)$         | 36s    |
> | $\mathcal{N}(\begin{bmatrix} 2 \\ 2 \end{bmatrix},0.01I)$          | 38s    |
> | $\mathcal{N}(\begin{bmatrix} -11 \\ -1 \end{bmatrix},0.5I)$        | 41     |
> | $\mathcal{N}(\begin{bmatrix} 11 \\ 1 \end{bmatrix},0.5I)$      | 36     |
> | GMM1   | 3m10s  |
> | $\mathcal{N}(\vec{0}_2,\text{diag}\begin{bmatrix} 0.5 \\ 0.25 \end{bmatrix})$     | 41s    |
> | $\mathcal{N}(\vec{0}_2,3I)$    | 41s    |
> | $\mathcal{N}(\vec{0}_{1000},\text{diag}\begin{bmatrix} 4 \\ 0.25 \\ \vdots \\ 0.25 \end{bmatrix})$ | 29m35s |
> | $\mathcal{N}(\vec{0}_{1000},3I)$     | 31m12s |
>
>  By examining the number of parameters, we noticed that PDPO has fewer parameters in most cases. To test whether parameter count explains PDPO's speed advantage, we solve S-Curve ($\sigma = 0$) and V-Neck ($\sigma = 1$) examples using identical MLP architectures [2+1,128,4] for both methods. Both problems use $\rho_0 = \mathcal{N}(\begin{bmatrix} -11 \\ -11 \end{bmatrix},0.5I)$ and $\rho_1 = \mathcal{N}(\begin{bmatrix} 11 \\ 11 \end{bmatrix},0.5I)$ with the same pretrained boundary parameters $\theta_0, \theta_1$ (training times: 0.33s and 0.35s respectively).
>
> | Problem | Method | Time | $\mathcal{A}$ | $W_2^2(t=0)$ | $W_2^2(t=1)$ |
>
> |---------|--------|------|-------------|--------------|--------------|
>
> | S-Curve | PDPO | 12m34s | 471.86 | 0.099 | 0.115 |
>
> | S-Curve | GSBM | 25m35s | 485.69 | 1.816 | 1.835 |
>
> | V-Neck | PDPO | 1h10m | 148.91 | 0.085 | 0.081 |
>
> | V-Neck | GSBM | 1h16m | 153.80 | 0.192 | 0.283 |
>
> With identical architectures, PDPO still outperforms GSBM in both action value and boundary approximation quality, demonstrating that performance differences are not due to parameter count alone. This also illustrates boundary parameter reusability across different problems.
>
>
> ## **Q2: Clarification on Constraint Violation in Figure 2**
>
> You're absolutely right - we need quantitative measures! Here's the breakdown of $\mathbb{E}_{X\sim\rho(t)}[V(X)]$ where $V$ is **only the obstacle potential**:
> | Method   |t = 0.1 |t = 0.2  | t = 0.3  | t = 0.4  |t = 0.5  |t = 0.6 | t = 0.7  | t = 0.8  | t = 0.9 | $\int_0^1\mathbb{E}_{X\sim\rho(t)}[V(X)]dt$  |
> | -------- | ------- | ------- | ------- | ------- | ------- | ------- | ------- | ------- | ------- | ----- |
> | PDPO     | 0.03 |0.72 | 3.21 | 0.0 | 0.0     | 0.0     | 4.31     | 0.0     | 0.0     | 0.99   |
> | GSBM     | 3.56 | 6.37 | 2.37 | 0.34$ | 0.4$ | 3.46 | 0.14      |    3.76     |    1.24     |  2.57     |
> | APAC-Net |   0.77  | 4.75    |    0.52     |    0.02     |    0.63     |    0.91     |     7.27    |    0.82     |    0.46     |   1.48    |
>
> This **quantitatively** shows that PDPO finds the path with less constraint violation.
>
> ## **Q3: Request for Concrete Implementation Details**
>
> **Parametrization of the Pushforward Map:**
>
> You've identified a fundamental paradigm mismatch that's worth clarifying. PDPO treats neural network parameters as function arguments in a functional programming style, while PyTorch follows an object-oriented paradigm where parameters are object attributes.
>
> The Core Challenge: When we interpolate spline control points $\theta(\tau_j)$ and need to evaluate the pushforward $(T_{\theta(\tau_j})\\#\lambda$, PyTorch's '''torch.load()''' operation treats $\theta(\tau_j)$ as static model states rather than dynamic function arguments. This breaks the computational graph connecting $\theta(\tau_j)$ back to the optimizable control points.
>
> Current Solution: We've implemented parametric MLPs that accept parameters as explicit function arguments:
>
> ```python
> def forward(self, x, theta):
>     return parametric_mlp_forward(x, theta)
> ```
> We acknowledge this currently restricts us to architectures we can reimplement from scratch. This is a genuine limitation for practitioners wanting to use existing complex architectures like U-Nets for image-based transport.
>
> Why JAX/Flax Resolves This: JAX's functional programming paradigm naturally aligns with our approach. In Flax, parameters are function arguments by design. This eliminates the computational graph issues entirely and allows direct use of any Flax architecture without implementation. We're actively developing the JAX/Flax implementation and expect to demonstrate arbitrary architecture support (including U-Net equivalents) in follow-up work.
>
>
> **Architecture details:**
>
> | Problem | Architecture | Meaning |
> |---------|-------------|---------|
> | V-neck | [2,128,4] | 2D input, 128 hidden units, 4 layers |
> | GMM | [2,256,4] | 2D input, 256 hidden units, 4 layers |
> | Opinion | [1000,128,4] | 1000D input, 128 hidden units, 4 layers |
>
> **Flow Matching Initialization step-by-step:**
>
> First, select the architecture for the NODE. For example, an MLP with 4 layers each with 128 neurons.
>
> Then, for each boundary condition:
> 1. Generate training dataset from target distribution
> 2. Train NODE using FM loss with a random coupling.
> 3. Store state dictionary: `torch.save(model.state_dict())`
>
> We recommend training boundary conditions $\theta_0$ and $\theta_1$ **in parallel** for efficiency.
>
> To initialize the PDPO algorithm:
> 1. Load trained networks: `torch.load('path_to_state_dictionary')`
> 2. Convert to tensor using the `state_dictionary_to_tensor` function for each network, obtaining the tensors $\theta_0$ and $\theta_1$
> 3. Initialize control points: $\theta_{t_i} = (1-i/K)\theta_0 + (i/(K+1)\theta_1.$
> 4. Using the torch tensors $\{\theta_{t_i}\}_{i = 0}^{K+1}$ use Algorithm 1 using parametric_mlps as described before.
>
>
> ## References
> [1] GSBM

---

> > ### Comment · Reviewer_5bmt · 2025-08-05
> > **Rebuttal Recieved**
> >
> > I would like to thank the authors for their comprehensive and thoughtful rebuttal. By providing new experimental results, including a crucial ablation study, and clarifying the justification for their design choices, they have successfully addressed the points raised in my initial review.

---

> > > ### Author Response · Authors · 2025-08-05
> > >
> > > Dear reviewer 5bmt. The authors thank you for taking the time to thoroughly review our rebuttal and for acknowledging that we successfully addressed the concerns raised in your initial review. We greatly appreciate your recognition of our comprehensive response, new experimental results, and clarifications.
> > > Given that you found our rebuttal satisfactory in addressing all the points from your initial review, we respectfully wonder if this might warrant reconsidering the numerical score to better reflect the resolution of the previously identified concerns. We understand that score adjustments are at the reviewer's discretion, but we wanted to express that such a change would be greatly appreciated and would help ensure the evaluation aligns with your positive assessment of our responses.
> > > Regardless, we sincerely thank you for your constructive feedback throughout this process, which has undoubtedly strengthened our work.

---

### Official Review · Reviewer_CE31 · 2025-07-05

**Clarity:** 2
**Significance:** 3
**Originality:** 3
**Rating:** 5
**Confidence:** 2

**Summary:**

This paper introduces a method for the optimisation of paths followed by probability distributions under trajectory constraints. The proposed approach transforms the original Wasserstein-2 optimal transport problem into an equivalent dynamic reinterpretation and formulates a constrained least-action optimisation objective. This dynamic formulation more easily allows for the incorporation of additional trajectory constraints. Probability distributions and their paths and their paths are represented by parametric models. Cubic splines parameterise the paths, with endpoint densities given by flow matching models to approximate the target distributions and intermediate points approximated via neural ODEs. Theoretical results regarding the equivalence of the formulations and the approximation error are derived. Experiments on synthetic benchmarks evaluate the algorithms against the most relevant baselines in the area.

**Questions:**

* The final algorithms have quite a few components and moving parts (flow matching models, neural ODEs, cubic spline parameters, Monte Carlo estimators for entropy, Fisher Information, etc.). How are hyper-parameters tuned? The appendix presents an ablation study for the number of control points, but the paper would benefit from a discussion on how sensitive the method would be to the choice of models involved and the machinery involved in the numerical integration and optimisation steps.

* With the spline-based framework, the method seems flexible enough to incorporate more than 2 fixed endpoints. Have the authors considered problems where intermediate points also need to satisfy hard constraints?

* Can you revise and explain the definition of $\tilde{\Pi}$?

**Ethical Concerns:**

["NO or VERY MINOR ethics concerns only"]

**Final Justification:**

I've increased my review score given that the rebuttals have addressed most of my concerns and other relevant points I found raised in other reviews. There are a few details to clarify in the revised paper, which should be taken seriously, given the feedback from the reviews, and I'd like encourage the authors to do so.

**Limitations:**

Limitations are quite briefly mentioned in the conclusion, without much detail or discussion.

**Paper Formatting Concerns:**

Some in-text citations need better formatting. For example, when referring directly to a paper, the authors should be named, i.e., "as in Author(s) [123]", instead of "as in [123]". The "\citet{}" command from natbib should do the appropriate formatting.

**Quality:**

3

**Strengths And Weaknesses:**

### Strengths

* The paper tackles a hard problem with an interesting solution that connects different fields.

* Theoretical guarantees are provided.

* The discussion of experimental results seems to be solid, insightful and to go into enough depth for a conference paper.

* Significant computational gains with competitive performance are demonstrated.


### Weaknesses

* The paper is quite mathematically heavy at times with notation that turns out a bit loosely defined and a few potential mistakes.

    - In the introduction and subsequent restatements, the action functional equation (1) has a minimisation over two function-valued variables $\rho$ and $v$, but they are indexed (under the $\inf$) as scalars representing their point evaluations $\rho(t, x)$ and $v(t, x)$, which is confusing.
    - How can a joint probability density $\pi$ be equated to a point tuple as $\pi = (x, T(_\pi(x))$? These are two objects of different nature. The same apparent mistake occurs first between lines 134 and 135 and it]s then repeated throughout the text. The set $\tilde{\Pi}$ they define then appears in Eq. 11, which is part of the result in Theorem 1, making the latter seem invalid.

* "Opinion" dynamics is involved in the experiments, but the term is never explained or defined in the main text.

* Quite a few of the references to the appendix are invalid in the main text, though fixed in the supplementary material.

---

> ### Author Rebuttal · Authors · 2025-07-31
>
> You're absolutely right about the notation problems. Let us clarify:
>
> ## **Equation (1) notation:**
>
>  We'll change $\rho(t,x), v(t,x)$ to simply $\rho, v$ under the infimum to be mathematically correct.
>
> ## **Definition of $\tilde{\Pi}$**
>
> We agree that the notation $\pi(x,y) = (x,T_{\pi}(x))$ is wrong. The reviewer is correct, and the definition of $\tilde{\Pi}$ needs updating, although **Theorem 1 keeps its validity** as we demonstrate below.
>
> To define the set of couplings properly, we first define the projections $p_x$ and $p_y$ as the projections of $\mathbb{R}^d \times \mathbb{R}^d$ onto the first and second $\mathbb{R}^d$ components, respectively. Given $z \in \mathbb{R}^d \times \mathbb{R}^d$ with $z = (x,y)$ where $x,y \in \mathbb{R}^d$, we have $p_x(z) = x$ and $p_y(z) = y$.
>
> Given probability measures $\mu_0, \mu_1$, the set of couplings between $\mu_0$ and $\mu_1$ is:
> $$\Pi(\mu_0,\mu_1) :=\\{\pi \in \mathcal{P}(\mathbb{R}^d \times \mathbb{R}^d): (p_x)\\#(\pi) = \mu_0, (p_y)\\#(\pi) = \mu_1\\}$$
>
> In our paper, we focus on couplings of the form $\pi = (id,T)\\#(\mu_0)$, where $T: \mathbb{R}^d \to \mathbb{R}^d$ is a measurable map such that $T\\#(\mu_0) = \mu_1$, and $id(\cdot)$ is the identity map: $id(x) = x$.
>
> We acknowledge that in the paper, we made an abuse of notation using measures and densities interchangeably. Here we provide the rigorous treatment: we use $\mu$ for measures and $\rho$ for probability densities.
>
> **Corrected definition of $\tilde{\Pi}$:**
> $$\tilde{\Pi} := \{\pi \in \Pi(\mu_0,\mu_1): \pi = (id,T_{\pi})\\#(\mu_0) \text{ for some injective measurable map } T_{\pi}: \mathbb{R}^d \to \mathbb{R}^d\}$$
>
> **Why this is well-defined:** The hypothesis that $\mu_0$ is absolutely continuous with respect to the Lebesgue measure allows us to use Brenier's theorem, which guarantees that the coupling $\pi^{\*}$ achieving optimal transportation cost lies on a graph. That is, $\pi^{\*} = (id,T^{\*})\\#(\mu_0)$ where $T^{\*}$ is the Monge map. Since we also assume $\mu_1$ is absolutely continuous, this map is bijective, ensuring $\tilde{\Pi} \neq \emptyset$.
>
> **Proof that Theorem 1 remains valid:**
>
> The optimization problems:
> $$\inf_{\rho,v} \int_0^1 \int_{\mathbb{R}^d} \frac{1}{2}\|v(t,x)\|^2 \rho(t,x) dx + F(\rho(t)) dt$$
> subject to $\partial_t \rho + \nabla \cdot (\rho v) = 0, \rho(0,\cdot) = \rho_0, \rho(1,\cdot) = \rho_1$ (dynamic problem)
>
> and
> $$\inf_{\pi \in \tilde{\Pi}(\rho_0,\rho_1)} c(\pi) = \inf_{\pi \in \tilde{\Pi}(\rho_0,\rho_1)} \inf_{\gamma \in \Gamma(\pi)} \mathcal{A}(\gamma)$$
> (static problem) are equivalent.
>
> **Direction 1 (Dynamic → Static):** A local minimizer $(\rho,v)$ of the dynamic problem defines the $C^1$-diffeomorphic curve:
> $$\gamma_{\rho,v}(t,x) = x + \int_0^t v(s,\gamma(s,x)) ds$$
> and coupling $\pi_{\rho,v} = (id,\gamma(1,\cdot))\\#(\mu_0)$.
>
> From the definition of $\gamma_{\rho,v}$, the distribution $\mu_{\gamma_{\rho,v}(t)} = (\gamma_{\rho,v}(t,\cdot))\\#(\mu_0)$ has density given by the change of variables formula.
> and $\rho_{\gamma_{\rho,v}(t)}(x) = \rho(t,x)$, thus:
> $$\int_0^1 \int_{\mathbb{R}^d} \frac{1}{2}\|v(t,x)\|^2 \rho(t,x) dx + F(\rho(t)) dt = \mathcal{A}(\gamma_{\rho,v})$$
>
> Therefore, the dynamic problem upper bounds the static problem.
>
> **Direction 2 (Static → Dynamic):** Given a local minimizer $(\pi,\gamma)$ of the static problem, let $\mu_{\pi,\gamma} = (\gamma_t(\cdot))\\#(\mu_0)$ with corresponding density $\rho_{\pi,\gamma}(t,\cdot)$. Define the velocity field:
> $$v_{\pi,\gamma}(t,\gamma(t,x)) = \frac{d}{dt}\gamma(t,x)$$
>
> The pair $(\rho_{\pi,\gamma}, v_{\pi,\gamma})$ satisfies the conditions of the dynamic problem. As before:
> $$\mathcal{A}(\gamma) = \int_0^1 \int_{\mathbb{R}^d} \frac{1}{2}\|v_{\pi,\gamma}(t,x)\|^2 \rho_{\pi,\gamma}(t,x) dx + F(\rho_{\pi,\gamma}(t)) dt$$
>
> Thus, the static problem upper bounds the dynamic problem.
>
> Since both problems upper-bound each other, they are equivalent
>
> ## **Extensions and Future Work**
>
> **Intermediate constraints:** We have thought about it, and plan to work on a PDPO extension for these type of problems. In that direction, we preview that a modification on how the coupling works in Theorem 1 will have to be made. This will also have an effect on how the boundary parameters (and fixed intermediate points) need to be updated.
>
> Other feasible directions for extending our work are:
> - "Overdamped Brownian Motion in a Force Field" see section 8 in [1]
> - "Density Control With Multiple Species", see section 4 [2]
> - Optimal transport with nonlinear diffusion equations, see [3]
>
>
> ## **Hyperparameter Sensitivity and Tuning**
>
> **Flow Matching Architecture:**
> The FM model defines the parameter space and must be sufficiently expressive to capture the boundary distributions. Key considerations:
> - **Architecture sizing**: Use larger networks than minimally required for boundary fitting. For example, while [2,64,4] can learn our V-neck boundaries accurately, we needed [2,128,4] to capture the solution complexity when potential terms significantly alter the density shape.
> - **Rule of thumb**: If the potential energy terms are expected to create new modes or dramatically change boundary shapes, increase the hidden dimension.
>
>
> **Neural ODE Integration:**
> Two factors affect *forward quality:
> - **Solver choice**: We use midpoint integration, as it is standar in flow based generative models.
> - **Integration steps**: 10 steps suffice for most problems. Use the maximum of the number of steps that your boundary NODEs require for accurate sampling from target datasets.
>
> **Action Approximation (Time Steps N):**
> This an important parameter.
>
> | N | Action | Kinetic Energy | Potential Energy |
> |---|--------|----------------|------------------|
> | 10 | 66.06 | 37.38 | 28.68 |
> | 20 | 42.09 | 37.77 | 12.82 |
> | 30 | 30.31 | 29.46 | 0.788 |
> | 40 | 30.20 | 29.72 | 0.735 |
> | 50 | 30.10 | 29.37 | 0.725 |
>
> **Recommendation**: Start with N=20 minimum, increase until action converges (typically N=30-50). The dramatic drop from N=10 to N=20 shows why sufficient discretization is crucial. This parameter is related to the complexity of the problem, the longer the distance and the more complex the landascape,
>
>
> **Monte Carlo samples M**: If the number of samples affects the quality of the estimate. Although even for the 50D case with 100,0 we were able to maintain the quality of the solution as shown in the Appendix
>
> **Computation of $\log(\rho)$ and $\nabla\log(\rho)$** The scaling issue of the computation of $\log(\rho)$ in the CNF paper [4], the Hutchinson trace estimator is used to make the ODE for $\log(\rho)$ computaionally tractable. To the best of our knowledge, there are no estamator techinques for $\nabla\log(\rho)$ and we consider this to be an interesting problem for future development.
>
>
> ## **Reference Formatting**
>
> We'll update citations to use proper author names: "as shown by Chen et al. [10]" instead of "as in [10]".
>
> ## References
> [1] On the Relation Between Optimal Transport and Schrödinger Bridges: A Stochastic Control Viewpoint Yongxin Chen, et all.
> [2]Density Control of Interacting Agent Systems, Yongxin Chen, IEEE TRANSACTIONS ON AUTOMATIC CONTROL, VOL. 69, NO. 1, JANUARY 2024.
>
> [3] Nonlinear mobility continuity equations and generalized  displacement convexity. J. A. Carrillo, S. Lisini, G. Savare , D. Slepcev. Journal of Functional Analysis Volume 258, Issue 4, 15 February 2010, Pages 1273-1309.
>
> [4] FFJORD: FREE-FORM CONTINUOUS DYNAMICS FOR SCALABLE REVERSIBLE GENERATIVE MODELS. Will Grathwohl et al.  ICLR 2019

---

> ### Comment · Reviewer_CE31 · 2025-08-05
>
> I'd like to thank the authors for addressing most of my concerns. I'd strongly encourage the authors to take into account the feedback from the reviews and rebuttal discussions, given that there seem to be quite a few details clarify. I'll update my score.

---

> ### Author Response · Authors · 2025-08-05
>
> We thank you for your constructive feedback throughout this process and for recognizing that we addressed your concerns.
>
> We will **certainly** incorporate the valuable feedback from all reviewers and rebuttal discussions into the final manuscript, including the mathematical clarifications, hyperparameter guidance, experiments, and additional technical details that emerged from our discussions.
>
> Your and others' reviewers' suggestions have strengthened the paper, and we're grateful for your time and expertise.

---

### Note · Authors · 2025-08-12

We sincerely thank the Area Chair and reviewers for their time, thoughtful feedback, and constructive discussions.

In our rebuttal, we addressed all questions in detail, providing new theoretical clarifications, additional experiments, and quantitative analyses. These exchanges have strengthened the paper, and in the final version, we will incorporate their suggestions with additional clarifications and numerical evidence from our rebuttal.

We note that Reviewer 5bmt explicitly stated our rebuttal was **comprehensive and thoughtful** and that we had **successfully addressed all their concerns**, yet their numerical score was not increased.

In the case of Reviewer Q47s, **several major criticisms were based on factual misunderstandings** (e.g., conflating theoretical and empirical claims, misreading Theorem 1’s assumptions, overlooking our fairness experiments, initialization analysis, and possible misunderstanding of NODE use). These were **explicitly corrected** in our rebuttal with **rigorous** arguments and empirical evidence, but the review was not adequately updated to reflect this by the claim of *substantial gaps in the original submission*, and **not** responding to our further clarifications.

---

### Decision · Program_Chairs · 2025-09-17

**Decision:**

Accept (poster)

**Comment:**

This paper presents a new way to find the most efficient paths between probability densities, under the theoretical framework of optimal transport. It converts an infinite-dimensional optimization into a finite dim over parameters, using cubic splines to map those paths. The approach handles complex scenarios like obstacles, mean-field effects, and stochastic controls, and it performs better than current top methods on benchmarks.
The proposed algorithm PDPO demonstrates clear improvements in runtime compared to existing methods.
As pointed out by a reviewer, the only baseline the authors claimed that can be compared with is GSBM. The usability of proposed method could be more convincing with comprehensive comparisons.
Three reviewers chose accept or borderline accept. One reviewer suggested borderline reject. After reading the paper and discussions, the AC tend to accept.